# FRUGALNERF: FAST CONVERGENCE FOR FEW-SHOT NOVEL VIEW SYNTHESIS WITHOUT LEARNED PRIORS

## ABSTRACT

Neural Radiance Fields (NeRF) face significant challenges in few-shot scenarios, particularly due to overfitting and long training times for high-fidelity rendering. While current approaches like FreeNeRF and SparseNeRF use frequency regularization or pre-trained priors, they can be limited by complex scheduling or potential biases. We introduce FrugalNeRF, a novel few-shot NeRF framework that leverages weight-sharing voxels across multiple scales to efficiently represent scene details. Our key contribution is a cross-scale geometric adaptation training scheme that selects pseudo ground truth depth based on reprojection error from both training and novel views across scales. This guides training without relying on externally learned priors, allowing FrugalNeRF to fully utilize available data. While not dependent on pre-trained priors, FrugalNeRF can optionally integrate them for enhanced quality without affecting convergence speed. Our method generalizes effectively across diverse scenes and converges more rapidly than state-of-the-art approaches. Our experiments on standard LLFF, DTU, and RealEstate-10K datasets demonstrate that FrugalNeRF outperforms existing few-shot NeRF models, including those using pre-trained priors, while significantly reducing training time, making it a practical solution for efficient and accurate 3D scene reconstruction.

## 1 INTRODUCTION

Few-shot novel view synthesis, generating new views from limited imagery, poses a substantial challenge in computer vision. While Neural Radiance Fields (NeRF) (Mildenhall et al., 2020) have revolutionized high-fidelity 3D scene recreation, they demand considerable computational resources and time, often relying on external datasets for pre-training. This paper introduces *FrugalNeRF*, a

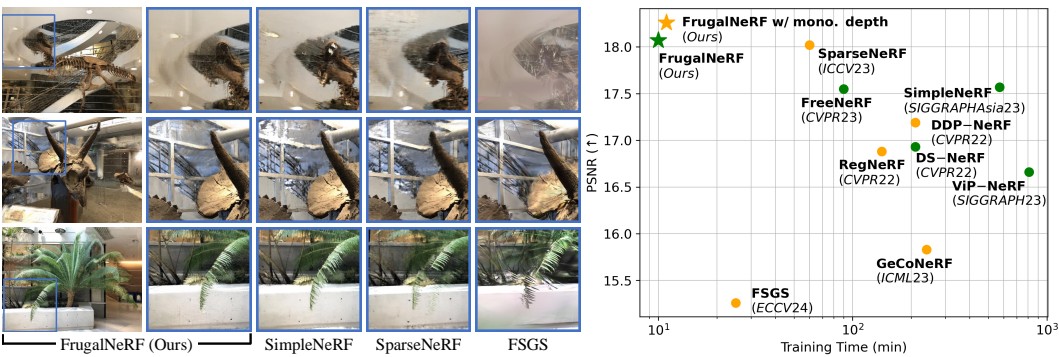

Figure 1: **Comparison of novel view synthesis methods trained on two views.** SimpleNeRF (Somraj et al., 2023) suffers from long training times, SparseNeRF (Wang et al., 2023) produces blurry results, and FSGS (Zhu et al., 2023) quality drops with few input views. Our FrugalNeRF achieves rapid, robust voxel training without learned priors, demonstrating superior efficiency and realistic synthesis. It can also integrate pre-trained priors for enhanced quality. **Green**: methods *without* learned priors. **Orange**: *with* learned priors

Figure 2: **Comparisons between few-shot NeRF approaches.** (a) Frequency regularization gradually increases the visibility of high-frequency signals of positional encoding, but the training speed is slow. (b) Replacing the MLPs with voxels and incorporating them with gradual voxel upsampling achieves similar frequency regularization but cannot generalize well. (c) Some approaches employ pre-trained models to supervise the rendered color or depth patches. (d) Our FrugalNeRF, leveraging weight-sharing voxels across scales for various frequencies representation, enhanced by a cross-scale geometric adaptation for efficient supervision.

novel approach to accelerate NeRF training in few-shot scenarios. It fully leverages the training data without relying on external priors and markedly reduces computational overhead.

Traditional NeRF methods, despite producing high-quality outputs, suffer from long training time and rely on frequency regularization (Yang et al., 2023) via multi-layer perceptrons (MLPs) and positional encoding, slowing convergence (Fig. 2 (a)). Alternatives like voxel upsampling (Fig. 2 (b)) attempt to overcome these challenges but struggle with generalizing to varied scenes (Chen et al., 2022a; Sun et al., 2022; 2023). Furthermore, using pre-trained models (Fig. 2 (c)) creates dependencies on external priors, which might not be readily available or could introduce biases from their training datasets (Niemeyer et al., 2022; Roessle et al., 2022; Wang et al., 2023).

FrugalNeRF differs from these approaches by incorporating a cross-scale, geometric adaptation mechanism, facilitating rapid training while preserving high-quality view synthesis (Fig. 2 (d)). Our method efficiently utilizes weight-sharing voxels across various scales to encapsulate the scene's frequency components. Our proposed adaptation scheme projects rendered depths and colors from different voxel scales onto the closest training view to compute reprojection errors. The most accurate scale becomes the pseudo-ground truth and guides the training across scales, thus eliminating the need for complex voxel upsampling schedules and enhancing generalizability across diverse scenes.

FrugalNeRF significantly reduces computational demands and accelerates training through self-adaptive mechanisms that exploit the multi-scale voxel structure, ensuring quick convergence without compromising the synthesis quality. By fully leveraging the training data and eliminating reliance on externally learned priors and their inherent limitations, FrugalNeRF provides a pathway toward more scalable and efficient few-shot novel view synthesis. In conclusion, FrugalNeRF efficiently bypasses the need for external pre-trained prior and complex scheduling for voxel.

We evaluate the FrugalNeRF's effectiveness on three prominent datasets: LLFF (Mildenhall et al., 2019b), DTU (Jensen et al., 2014), and RealEstate-10K Zhou et al. (2018) dataset to assess both the rendering quality and convergence speed. Our results show that FrugalNeRF is not only faster but also achieves superior quality in comparison to existing methods (Fig. 1), showcasing FrugalNeRF's proficiency in generating perceptually high-quality images. The main contributions of our work are:

- We introduce a novel weight-sharing voxel representation that encodes multiple frequency components of the scene, significantly enhancing the efficiency and quality of few-shot novel view synthesis.

- Our geometric adaptation selects accurate rendered depth across different scales by reprojection errors to create pseudo geometric ground truth that guides the training process, enabling a robust learning mechanism that is less reliant on complex scheduling and more adaptable to various scenes.

- FrugalNeRF's training scheme relies solely on available data, eliminating the need for external priors or pre-trained models and ensuring fast convergence without sacrificing quality. It remains flexible, allowing the integration of learned priors to further enhance quality without affecting training speed.

## 2 RELATED WORK

Neural Radiance Fields (NeRF)(Mildenhall et al., 2020) has advanced novel view synthesis(Chen et al., 2022c; Martin-Brualla et al., 2021; Yuan et al., 2022; Xu et al., 2023; Tao et al., 2023; Chen et al., 2023; Peng et al., 2021; Xu et al., 2022; Fridovich-Keil et al., 2022; Zhang et al., 2020; Wang et al., 2021b; Ye et al., 2022; Zheng et al., 2023; Bian et al., 2023). Research spans multi-view synthesis (Oechsle et al., 2021; Chen et al., 2021; Jensen et al., 2014; Yariv et al., 2020; Wang et al., 2021a), single-view synthesis (Gao et al., 2020; Tucker & Snavely, 2020; Han et al., 2022; Wiles et al., 2020; Wimbauer et al., 2023), 3D generation (Chan et al., 2021; Wang & Torr, 2022; Chan et al., 2022; Hong et al., 2023; Li et al., 2021), and dynamic scenes (Pumarola et al., 2021; Mildenhall et al., 2022; Liu et al., 2023). Few-shot NeRFs (Chibane et al., 2021; Hu et al., 2023a;b; Chen et al., 2022b; Zhang et al., 2021; Jain et al., 2022; Zhou & Tulsiani, 2023; Kim et al., 2022; Bortolon et al., 2022; Lee et al., 2023; Seo et al., 2023b; Kwak et al., 2023) aim to reconstruct from sparse inputs but face overfitting and generalization issues. Some approaches use pre-trained models (Yu et al., 2021; Jain et al., 2021; Wang et al., 2023; Niemeyer et al., 2022; Johari et al., 2022; Deng et al., 2023; Chen et al., 2016; Uy et al., 2023), while others introduce regularizations (Yang et al., 2023; Niemeyer et al., 2022; Somraj & Soundararajan, 2023; Deng et al., 2022) to improve performance.

**Depth regularizations.** Recent works emphasize depth constraints during training. DS-NeRF (Deng et al., 2022) uses sparse SfM-estimated depth, while DDP-NeRF (Roessle et al., 2022) completes it with pretrained priors. SparseNeRF (Wang et al., 2023) uses prediction transformers (Ranftl et al., 2021a; 2020) for depth priors. DäRF (Song et al., 2023) jointly optimizes NeRF and MDE, and ReVoRF (Xu et al., 2024) improves geometry without heavy reliance on priors. FSGS (Zhu et al., 2023) uses monocular depth priors and geometric regularization. These methods may be affected by data bias and require substantial data. ViP-NeRF (Somraj & Soundararajan, 2023) uses visibility maps but demands significant computation time. In contrast, FrugalNeRF uses geometrically adapted pseudo-GT depth, avoiding pre-trained models and extensive computation.

**Novel pose regularization.** Novel pose regularization addresses floaters in synthesized views from sparse inputs. RegNeRF (Niemeyer et al., 2022) uses pose sampling with a normalizing flow model. PixelNeRF (Yu et al., 2021) extracts image features with CNNs (Krizhevsky et al., 2012) for scene priors. DietNeRF (Jain et al., 2021) uses CLIP-based Transformers (Radford et al., 2021; Caron et al., 2021; Li et al., 2022; Lin et al., 2023) for color consistency. FlipNeRF (Seo et al., 2023a) samples flipped reflection rays but relies on geometry estimation. These methods often depend on pre-trained models, potentially introducing bias and inference time. Our FrugalNeRF applies geometric adaptation on pose rendering, avoiding pre-trained models while suppressing floaters.

**Frequency regularization.** Positional encoding (Sitzmann et al., 2020; Tancik et al., 2020; Wang et al., 2022) enables NeRF to capture high-frequency details but can lead to overfitting in few-shot scenarios. FreeNeRF (Yang et al., 2023) uses scheduling for increasing input frequency. VGOS (Sun et al., 2023) adopts incremental voxel training to prevent overfitting. Both methods require complex scheduling and may not generalize well. SimpleNeRF (Somraj et al., 2023) introduces augmented models focusing on low-frequency, leading to resource wastage. Our FrugalNeRF leverages weight-sharing voxels across scales for various frequency representations, avoiding complex scheduling.

**Fast convergence.** NeRF's time-consuming training due to MLP queries is a common challenge. Methods like (Sun et al., 2023; 2022; Chen et al., 2022a; Sitzmann et al., 2019) replace MLPs with faster-converging representations. Instant-NGP (Müller et al., 2022) uses voxels with hash encoding and density bitfield. DVGO (Sun et al., 2022) employs voxel grids with shallow MLP. TensoRF (Chen et al., 2022a) decomposes radiance fields into low-rank tensors. ZeroRF (Shi et al., 2024) adapts TensoRF for few-shot settings but is limited to the object level. Our FrugalNeRF uses TensoRF for fast training and introduces a cross-scale geometric adaptation weight-sharing voxel framework.

**Self-supervised consistency.** Consistency modeling between sparse images and warped counterparts is crucial for Few-shot NeRFs. SinNeRF (Xu et al., 2022) and PANeRF (Ahn et al., 2022) use warping results as pseudo labels but require RGB-D input. SE-NeRF (Jung et al., 2023) and Self-NeRF (Bai et al., 2023) use teacher NeRF rendering results as labels, requiring effective initialization. GeCoNeRF (Kwak et al., 2023) uses render depth for warping but needs a pre-trained feature extractor. FrugalNeRF combines frequency regularization with cross-scale geometric adaptation, using the best render depth at different scales as a pseudo label to ensure geometric consistency without relying on learned priors.

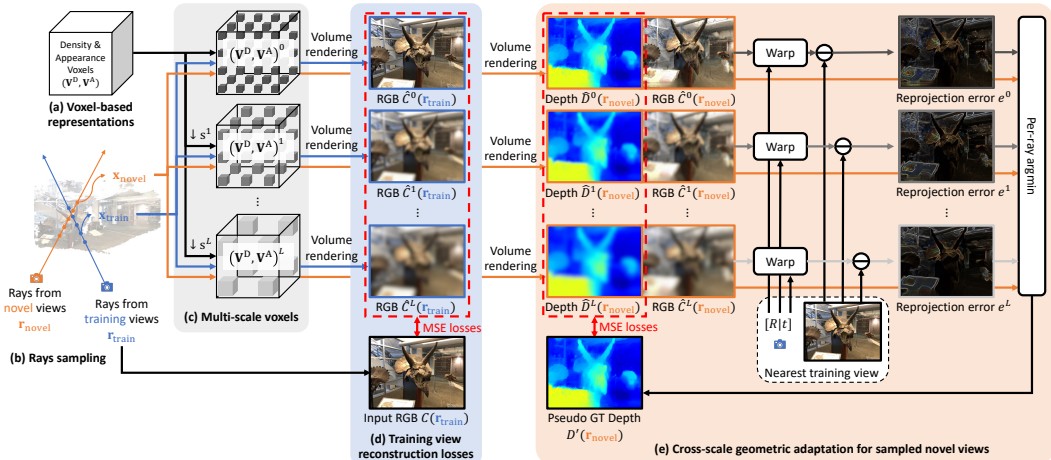

Figure 3: (a) Our FrugalNeRF represents a scene with a pair of density and appearance voxels $(\mathbf{V}^D, \mathbf{V}^A)$. For a better graphical illustration, we show only one voxel in the figure. (b) We sample rays from not only training input views $\mathbf{r}_{\text{train}}$ but also randomly sampled novel views $\mathbf{r}_{\text{novel}}$. (c) We then create $L + 1$ multi-scale voxels by hierarchical subsampling, where lower-resolution voxels ensure global geometry consistency and reduce overfitting but suffer from representing detailed structures, while higher-resolution voxels capture fine details but may get stuck in the local minimum or generate floaters. (d) For the rays from training views $\mathbf{r}_{\text{train}}$, we enforce an MSE reconstruction loss between the volume rendered RGB color $\hat{C}^l$ and input RGB $C$ at each scale. (e) We introduce a cross-scale geometric adaptation loss for novel view rays $\mathbf{r}_{\text{novel}}$, warping volume-rendered RGB to the nearest training view using predicted depth, calculating projection errors $e^l$ at each scale, and using the depth with the minimum reprojection error as pseudo-GT for depth supervision. This adaptation involves rays from both training and novel views, though the figure only depicts novel view rays for clarity.

## 3 METHOD

### 3.1 PRELIMINARIES

**Neural radiance fields.** NeRF (Mildenhall et al., 2020) uses a neural network $f$ to map 3D location $\mathbf{x}$ and viewing direction $\mathbf{d}$ to density $\sigma$ and color $\mathbf{c}$ for image rendering: $f : (\mathbf{x}, \mathbf{d}) \rightarrow (\sigma, \mathbf{c})$. Then we use the densities and colors to render a pixel color $\hat{C}(\mathbf{r})$ by integrating the contributions along a ray $\mathbf{r}$ cast through the scene: $\hat{C}(\mathbf{r}) = \sum_{i=1}^{N} T_i (1 - \exp(-\sigma_i \delta_i)) \mathbf{c}_i$, where $T(t) = \exp(-\sum_{j=i}^{i-1} \sigma_j \delta_j)$ is the transmittance along the ray, and $N$ is the number of points along the ray. NeRF seeks to minimize the MSE between the rendered image and the actual image: $\mathcal{L} = \sum_{\mathbf{r} \in \mathcal{R}} \left\| \hat{C}(\mathbf{r}) - C(\mathbf{r}) \right\|^2$, where $\mathcal{R}$ denotes a set of rays.

**Voxel-based NeRFs.** Voxel-based NeRFs (Sun et al., 2022; Chen et al., 2022a; Müller et al., 2022) enhance color and density querying speed in the radiance field by employing voxel grids, allowing efficient data retrieval via trilinear interpolation. They typically utilize a logistic function with a bias term for density calculation and adopt a coarse-to-fine strategy, refining results with a shallow MLP for view-dependent effects.

**Few-shot NeRFs.** Recent methods propose various strategies to address the challenge of under-constrained optimization with limited images. These include regularizing visible frequencies in positional encoding (Yang et al., 2023) (Fig. 2 (a)), expanding voxel ranges incrementally (Sun et al., 2023) (Fig. 2 (b)), and utilizing external priors like pre-trained models for additional guidance (Wang et al., 2023) (Fig. 2 (c)). Our approach, FrugalNeRF, leverages a weight-sharing voxel across scales to capture a spectrum of frequency components. It self-adapts by evaluating reprojection errors with the nearest training view, enhancing scene generalization, and offering faster training without dependence on pre-trained models (Fig. 2 (d)).

## 3.2 OVERVIEW OF FRUGALNERF

FrugalNeRF introduces an efficient architecture for novel view synthesis from sparse inputs without external priors, leveraging voxel-based NeRFs (Chen et al., 2022a; Müller et al., 2022; Sun et al., 2022) to estimate 3D geometry and reduce training time. Key features include hierarchical subsampling with weight-sharing multi-scale voxels for diverse geometric details (Sec. 3.3), a geometric adaptation training strategy for few-shot scenarios (Sec. 3.4), novel view sampling with additional regularization losses to minimize artifacts (Sec. 3.5), and integration of data from both training and sampled novel views for robust scene representation (Sec. 3.6).

## 3.3 WEIGHT-SHARING MULTI-SCALE VOXELS

Addressing data sparsity in few-shot scenarios, we introduce FrugalNeRF's weight-sharing multi-scale voxels, which are crucial for balancing frequency characteristics. Inspired by FreeNeRF (Yang et al., 2023), which highlights the overfitting challenges with high-frequency inputs, our system adopts a voxel-based representation to manage frequency components. We employ varied resolution voxels similar to NeRF's positional encoding (Mildenhall et al., 2020), with lower resolutions capturing broad scene outlines and higher resolutions modeling finer details.

Unlike methods such as VGOS (Sun et al., 2023), which starts with a coarse geometry and progressively refines details, our approach maintains generalization without intricate tuning. We construct multi-scale voxels by downsampling from a single density and appearance voxel, ensuring consistent scene representation(Fig. 3 (c)). This technique effectively balances different frequency bands in the training pipeline without increasing model size or memory demands.

With multi-scale voxels, we can further utilize *multi-scale voxel color loss* to guide the training (Fig. 3 (d)), which is crucial for few-shot scenarios in ensuring a balanced representation of geometry and detail. The multi-scale voxel color loss is defined as:

$$\mathcal{L}_{\text{ms-color}} = \sum_{l=0}^{L} \sum_{\mathbf{r}_{\text{train}} \in \mathcal{R}_{\text{train}}} \left\| \hat{C}^l(\mathbf{r}_{\text{train}}) - C(\mathbf{r}_{\text{train}}) \right\|^2, \tag{1}$$

where $\hat{C}^l$ is the rendered color from the voxel at scale $l$, $C$ is the ground truth color, $L$ is the number of scales, $\mathcal{R}_{\text{train}}$ is a set of rays from training views, and $\mathbf{r}_{\text{train}}$ is a ray sampled from $\mathcal{R}_{\text{train}}$. We compute a weighted average MSE loss across scales to ensure color rendering accuracy at each scale, enhancing overall robustness and fidelity.

## 3.4 CROSS-SCALE GEOMETRIC ADAPTATION

Our *cross-scale geometric adaptation* approach effectively addresses the challenges of few-shot scenarios by supervising geometry without ground truth depth data. Recognizing the diverse frequency representation by different voxel scales in a scene, it is essential to identify the optimal frequency band for each region of the scene.

For each ray from a training view $i$, we compute depth values at multiple scales through volume rendering and then warp (Luo et al., 2020; Kopf et al., 2021; Li et al., 2021) view $i$'s input RGB to the nearest training view $j$ using these depths. The reprojection error with view $j$'s input RGB determines the most suitable scale for each scene area. The depth of this scale serves as a pseudo-ground truth, guiding the model in maintaining geometric accuracy across frequencies (Fig. 3 (e)).

Mathematically, for a pixel $\mathbf{p}_i$ in a training frame $i$, with its depth $D_i^l(\mathbf{p}_i)$ at scale $l$ and camera intrinsic $K_i$, we can lift $\mathbf{p}_i$ to a 3D point $\mathbf{x}_i^l$, then transform it to world coordinate $\mathbf{x}^l$, and subsequently transform to frame $j$'s camera coordinate $\mathbf{x}_{i \to j}^l$. This 3D point is then projected back to 2D in frame $j$, obtaining the pixel coordinate $\mathbf{p}_{i \to j}^l$. Due to the space limit, we provide the details for reprojection calculation in the supplementary. We calculate the reproject error $e^l(\mathbf{p}_i)$ using the RGB values of frame $i$ and $j$ for each scale $l$.

$$e^l(\mathbf{p}_i) = \left\| C_i(\mathbf{p}_i) - C_j(\mathbf{p}_{i \to j}^l) \right\|^2, \tag{2}$$

where $C_i$ and $C_j$ are the input RGB images from view $i$ and $j$, respectively. For a pixel location $\mathbf{p}$ from which the training view ray $\mathbf{r}_{\text{train}}$ originates, we denote it simply as $\mathbf{r}_{\text{train}}$. The pseudo-ground truth depth for this pixel is the depth at the scale with the minimum reprojection error:

$$D'(\mathbf{r}_{\text{train}}) = \hat{D}^{l'(\mathbf{r}_{\text{train}})}(\mathbf{r}_{\text{train}}), \tag{3}$$

where $\hat{D}^l$ is the rendered depth from the voxel at scale $l$, and $l'$ denotes the scale with minimum reprojection error:

$$l'(\mathbf{r}_{\text{train}}) = \arg\min_l(e^l(\mathbf{r}_{\text{train}})). \tag{4}$$

This pseudo-ground truth depth $D'$ is used to compute a geometric adaptation loss, $\mathcal{L}_{\text{geo}}(\mathbf{r}_{\text{train}})$, an MSE loss that ensures the model maintains scene geometry effectively, even without explicit depth ground truth:

$$\mathcal{L}_{\text{geo}}(\mathbf{r}_{\text{train}}) = \sum_{l=0}^{L} \sum_{\mathbf{r}_{\text{train}} \in \mathcal{R}_{\text{train}}} \left\| \hat{D}^l(\mathbf{r}_{\text{train}}) - D'(\mathbf{r}_{\text{train}}) \right\|^2. \tag{5}$$

We further define a threshold for reprojection error to determine the reliability of depth estimation. Specifically, we do not compute the loss of those pixels in which the projection error exceeds this pre-defined threshold. Geometric adaptation is critical by allowing the model to refine its understanding of the scene's geometry in a self-adaptive manner.

### 3.5 Novel View Regularizations

In few-shot scenarios, we extend geometric adaptation to *novel views* to address the limitations in areas with less overlap among training views (Fig. 3 (e)). Our novel view sampling strategy involves a spiral trajectory around training views, promoting comprehensive coverage and model robustness. In the absence of ground truth RGB for novel views, we rely on rendered color $\hat{C}$ for reprojection error calculation, similar to Eq. (2) in Sec. 3.4, but focusing on rays from novel views $\mathbf{r}_{\text{novel}}$:

$$e^l(\mathbf{p}_n) = \left\| \hat{C}_n(\mathbf{p}_n) - C_j(\mathbf{p}_{n \to j}^l) \right\|^2. \tag{6}$$

In this context, $\mathbf{p}_n$ denotes a pixel coordinate in the sampled novel frame $n$, and $\mathbf{p}_{n \to j}^l$ represents the coordinates on its nearest training pose $j$ after warping $\mathbf{p}_n$ at scale $l$. This reprojection error helps refine the model's rendering for novel views. For each ray from a novel view, similar to Eqs. (3) to (5), we first determine the scale with the minimum reprojection error, then determine its pseudo-ground truth depth and calculate geometric adaptation loss:

$$l'(\mathbf{r}_{\text{novel}}) = \arg\min_l(e^l(\mathbf{r}_{\text{novel}})), D'(\mathbf{r}_{\text{novel}}) = \hat{D}^{l'(\mathbf{r}_{\text{novel}})}(\mathbf{r}_{\text{novel}}), \tag{7}$$

$$\mathcal{L}_{\text{geo}}(\mathbf{r}_{\text{novel}}) = \sum_{l=0}^{L} \sum_{\mathbf{r}_{\text{novel}} \in \mathcal{R}_{\text{novel}}} \left\| \hat{D}^l(\mathbf{r}_{\text{novel}}) - D'(\mathbf{r}_{\text{novel}}) \right\|^2, \tag{8}$$

where $\mathcal{R}_{\text{novel}}$ is the set of rays from sampled novel views, and $\mathbf{r}_{\text{novel}}$ is a sampled ray from the set $\mathcal{R}_{\text{novel}}$. We combine this loss with the geometric adaptation loss from training views to enhance the overall training process:

$$\mathcal{L}_{\text{geo}} = \mathcal{L}_{\text{geo}}(\mathbf{r}_{\text{train}}) + \mathcal{L}_{\text{geo}}(\mathbf{r}_{\text{novel}}). \tag{9}$$

This approach of novel view sampling and applying regularization through reprojection error computation is critical in training our model. It ensures that the model not only learns from the limited training views but also adapts to and accurately renders novel perspectives, thereby enhancing the overall performance and reliability of FrugalNeRF.

**Additional global regularization losses.** To further improve the geometry and reduce artifacts, we introduce an additional global regularization loss $\mathcal{L}_{\text{reg}}$, including total variation loss (Chen et al., 2022a; Sun et al., 2023), patch-wise depth smoothness loss (Niemeyer et al., 2022), L1 sparsity loss (Chen et al., 2022a), and distortion loss (Sun et al., 2022; Barron et al., 2022). These losses help smooth the scene globally and suppress artifacts like floaters and background collapse.

### 3.6 Total Loss

The total loss for FrugalNeRF, essential for accurate scene rendering from sparse views, combines various components: color fidelity, geometric adaptation, global regularization, and sparse depth constraints. It is formulated as:

$$\mathcal{L} = \mathcal{L}_{\text{ms-color}} + \lambda_{\text{geo}}\mathcal{L}_{\text{geo}} + \lambda_{\text{reg}}\mathcal{L}_{\text{reg}} + \lambda_{\text{sd}}\mathcal{L}_{\text{sd}}. \tag{10}$$

$\mathcal{L}_{\text{ms-color}}$ is the multi-scale voxel color loss, crucial for maintaining color accuracy across different scales. $\mathcal{L}_{\text{geo}}$ is the geometric adaptation loss, providing geometric guidance in the absence of explicit

Table 1: **Quantitative results on LLFF dataset (Mildenhall et al., 2019b).** FrugalNeRF performs competitively with baseline methods in extreme few-shot settings, offering shorter training time without relying on externally learned priors. Integrating monocular depth regularization further improves quality while maintaining fast convergence. Results differ from SimpleNeRF's paper but match its supplementary document, as we evaluate full images without visibility masks.

| Method | Venue | Learned priors | 2-view PSNR ↑ | SSIM ↑ | LPIPS ↓ | 3-view PSNR ↑ | SSIM ↑ | LPIPS ↓ | 4-view PSNR ↑ | SSIM ↑ | LPIPS ↓ | Training time ↓ |
|---|---|---|---|---|---|---|---|---|---|---|---|---|
| DS-NeRF (Deng et al., 2022) | CVPR22 | - | 16.93 | 0.51 | 0.42 | 18.97 | 0.58 | 0.36 | 20.07 | 0.61 | 0.34 | 3.5 hrs |
| FreeNeRF (Yang et al., 2023) | CVPR23 | - | 17.55 | 0.54 | 0.38 | 19.30 | 0.60 | 0.34 | 20.45 | 0.63 | 0.33 | 1.5 hrs |
| ViP-NeRF (Somraj & Soundararajan, 2023) | SIGGRAPH23 | - | 16.66 | 0.52 | 0.37 | 18.89 | 0.59 | 0.34 | 19.34 | 0.62 | 0.32 | 13.5 hrs |
| SimpleNeRF (Somraj et al., 2023) | SIGGRAPH Asia23 | - | 17.57 | 0.55 | 0.39 | 19.47 | 0.62 | 0.33 | 20.44 | 0.65 | 0.31 | 9.5 hrs |
| FrugalNeRF (Ours) | - | - | 18.07 | 0.54 | 0.35 | 19.66 | 0.61 | 0.30 | 20.70 | 0.65 | 0.28 | 10 mins |
| RegNeRF (Niemeyer et al., 2022) | CVPR22 | normalizing flow | 16.88 | 0.49 | 0.43 | 18.65 | 0.57 | 0.36 | 19.89 | 0.62 | 0.32 | 2.35 hrs |
| DDP-NeRF (Roessle et al., 2022) | CVPR22 | depth completion | 17.19 | 0.54 | 0.39 | 17.71 | 0.56 | 0.39 | 19.19 | 0.61 | 0.35 | 3.5 hrs |
| GeCoNeRF (Kwak et al., 2023) | ICML23 | VGG19 feature | 15.83 | 0.45 | 0.52 | 17.44 | 0.50 | 0.47 | 19.14 | 0.56 | 0.42 | 4 hrs |
| SparseNeRF (Wang et al., 2023) | ICCV23 | monocular depth | 18.02 | 0.52 | 0.45 | 19.52 | 0.59 | 0.37 | 20.89 | 0.65 | 0.34 | 1 hrs |
| FSGS (Zhu et al., 2023) | ECCV24 | monocular depth | 15.26 | 0.45 | 0.41 | 19.21 | 0.61 | 0.30 | 20.07 | 0.66 | 0.22 | 25 mins |
| FrugalNeRF (Ours) | - | monocular depth | 18.26 | 0.55 | 0.35 | 19.87 | 0.61 | 0.30 | 20.89 | 0.66 | 0.26 | 11 mins |

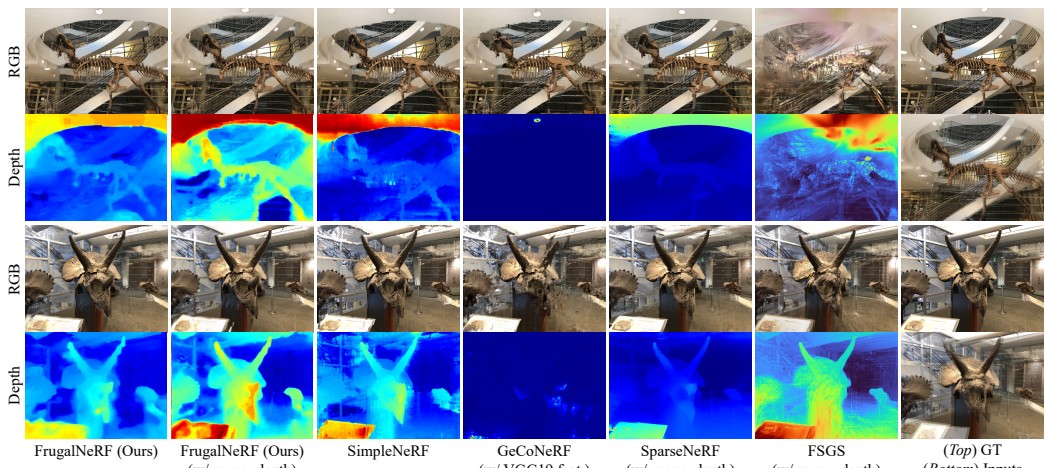

FrugalNeRF (Ours)    FrugalNeRF (Ours) (w/ mono. depth)    SimpleNeRF    GeCoNeRF (w/ VGG19 feat.)    SparseNeRF (w/ mono. depth)    FSGS (w/ mono. depth)    (*Top*) GT (*Bottom*) Inputs

Figure 4: **Qualitative comparisons on the LLFF (Mildenhall et al., 2019b) dataset with two input views.** FrugalNeRF achieves better synthesis quality and coherent geometric depth. We also include the GT and overlapped input images for reference.

depth information. $\mathcal{L}_{\text{reg}}$ is the global regularization loss, addressing artifacts and inconsistencies in unseen areas. And $\mathcal{L}_{\text{sd}}$ is the sparse depth loss (Deng et al., 2022), utilizing sparse depth data for absolute scale constraints derived from COLMAP (Schönberger & Frahm, 2016; Schönberger et al., 2016).

## 4 EXPERIMENTS

**Datasets & evaluation metrics.** We conduct experiments on two datasets: LLFF (Mildenhall et al., 2019b) , DTU (Jensen et al., 2014), and RealEstate-10K Zhou et al. (2018). For both datasets, we use the test sets defined by pixelNeRF (Yu et al., 2021) and ViP-NeRF (Somraj & Soundararajan, 2023). We follow the same evaluation protocol as ViP-NeRF, including the train/test split. Specifically, there are 12 scenes[1] in the test sets of the DTU dataset. We assume that camera parameters are known, which is relevant for applications with available calibrated cameras. We provide further details and RealEstate-10K in the supplementary materials.

We follow the established evaluation protocols for consistency. The experiments utilize three evaluation metrics: PSNR, SSIM (Wang et al., 2004), and LPIPS (Zhang et al., 2018). While evaluating on DTU, we follow SparseNeRF (Yang et al., 2023) to remove the background when computing metrics to alleviate the background bias reported by RegNeRF (Niemeyer et al., 2022) and pixelNeRF (Yu et al., 2021). Additionally, we include the training time with a single NVIDIA RTX 4090 GPU to evaluate the efficiency of the methods.

---

[1]There are 15 scenes in total in ViP-NeRF's DTU test sets. However, COLMAP can only run successfully on 12 scenes.

Table 2: **Quantitative results on the DTU Jensen et al. (2014) dataset.** FurgalNeRF synthesizes better images than most of the other baselines under extreme few-shot settings but with shorter training time and does not rely on any externally learned priors. Additionally, integrating monocular depth model regularization further improves quality while maintaining fast convergence. We follow SparseNeRF Wang et al. (2023) to remove the background when computing metrics.

| Method | Venue | Learned priors | 2-view PSNR ↑ | SSIM ↑ | LPIPS ↓ | 3-view PSNR ↑ | SSIM ↑ | LPIPS ↓ | 4-view PSNR ↑ | SSIM ↑ | LPIPS ↓ | Training time ↓ |
|---|---|---|---|---|---|---|---|---|---|---|---|---|
| FreeNeRF Yang et al. (2023) | CVPR23 | - | 18.05 | 0.73 | 0.22 | 22.40 | 0.82 | 0.14 | 24.98 | 0.86 | 0.12 | 1 hrs |
| ViP-NeRF Somraj & Soundararajan (2023) | SIGGRAPH23 | - | 14.91 | 0.49 | 0.24 | 16.62 | 0.55 | 0.22 | 17.64 | 0.57 | 0.21 | 2.2 hrs |
| SimpleNeRF Somraj et al. (2023) | SIGGRAPH Asia23 | - | 14.41 | 0.79 | 0.25 | 14.01 | 0.77 | 0.25 | 13.90 | 0.78 | 0.26 | 1.38 hrs |
| ZeroRF Shi et al. (2024) | CVPR24 | - | 14.84 | 0.60 | 0.30 | 14.47 | 0.61 | 0.31 | 15.73 | 0.67 | 0.28 | 25 mins |
| FrugalNeRF (Ours) | - | - | 19.72 | 0.78 | 0.16 | 22.43 | 0.83 | 0.14 | 24.51 | 0.86 | 0.12 | 6 mins |
| RegNeRF Niemeyer et al. (2022) | CVPR22 | normalizing flow | - | - | - | - | - | - | - | - | - | OOM |
| SparseNeRF Wang et al. (2023) | ICCV23 | monocular depth | 19.83 | 0.75 | 0.20 | 22.47 | 0.83 | 0.14 | 24.03 | 0.86 | 0.12 | 30 mins |
| FSGS Zhu et al. (2023) | ECCV24 | monocular depth | 16.82 | 0.64 | 0.27 | 18.29 | 0.69 | 0.21 | 20.08 | 0.75 | 0.16 | 20 mins |
| FrugalNeRF (Ours) | - | monocular depth | 20.77 | 0.79 | 0.15 | 22.84 | 0.83 | 0.13 | 24.81 | 0.86 | 0.12 | 7 mins |

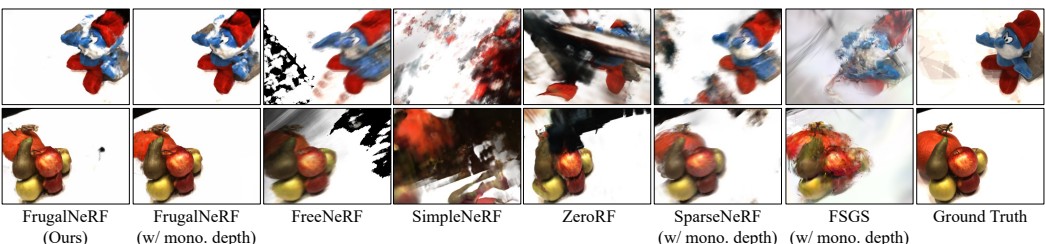

| FrugalNeRF (Ours) | FrugalNeRF (w/ mono. depth) | FreeNeRF | SimpleNeRF | ZeroRF | SparseNeRF (w/ mono. depth) | FSGS (w/ mono. depth) | Ground Truth |

Figure 5: **Qualitative comparisons on the DTU (Jensen et al., 2014) dataset with two input views.** FrugalNeRF achieves better synthesis quality.

**Implementation details.** We implement FrugalNeRF based on the TensoRF (Chen et al., 2022a) and utilize the official PyTorch framework. The learning process is driven by an Adam optimizer (Kingma & Ba, 2014), with an initial learning rate of 0.08, which decays to 0.002 throughout the training. We sample 120 novel poses along a spiraling trajectory around the training view and set the batch size for both training and novel view rays to 4,096. We utilize the pre-trained Dense Prediction Transformer (DPT) (Ranftl et al., 2021b) to generate monocular depth maps from training views. Each scene in our model is trained for 5,000 iterations. For different datasets, we use specific voxel resolutions: $640^3$ for LLFF and RealEstate-10K, and $300^3$ for the DTU dataset. Additionally, our model employs a voxel downsample ratio with $s = 4$, $L = 2$ (three levels of scale in total) to accommodate varying levels of scene detail. More details can be found in the supplementary materials.

### 4.1 COMPARISONS

**LLFF dataset.** We compare FrugalNeRF to RegNeRF (Niemeyer et al., 2022), DS-NeRF (Deng et al., 2022), DDP-NeRF (Roessle et al., 2022), FreeNeRF (Yang et al., 2023), ViP-NeRF (Somraj & Soundararajan, 2023), SimpleNeRF (Somraj et al., 2023), GeCoNeRF[2] (Kwak et al., 2023), SparseNeRF (Wang et al., 2023), and FSGS (Zhu et al., 2023). Some use pre-trained models or frequency regularization. As shown in Tab. 1, FrugalNeRF outperforms these methods in PSNR and LPIPS, with comparable SSIM. Our cross-scale geometric adaptation generalizes better than frequency regularization methods like FreeNeRF. Integrating monocular depth regularization further improves quality while maintaining fast convergence. FrugalNeRF achieves an optimal balance between quality and training time (10 minutes). Qualitative comparisons (Fig. 4) show that FrugalNeRF renders scenes with richer detail and sharper edges compared to SparseNeRF's blurry results. FrugalNeRF models scene geometry more smoothly and consistently than SimpleNeRF and FSGS, which suffer from floaters and holes. These results demonstrate FrugalNeRF's capability to model complex scenes with high fidelity.

**DTU dataset.** We compare FrugalNeRF with RegNeRF[3] (Niemeyer et al., 2022), FreeNeRF (Yang et al., 2023), ViP-NeRF (Somraj & Soundararajan, 2023), SimpleNeRF (Somraj et al., 2023),

---

[2]Since GeCoNeRF does not release a complete and executable implementation, we try our best to modify their code and reproduce its results.

[3]RegNeRF runs into an out-of-memory issue on one NVIDIA RTX 4090 GPU, so we cannot report its results on the DTU dataset

Table 3: **Comparison of different scales on the LLFF dataset.**

| # of scales | PSNR ↑ | SSIM ↑ | LPIPS ↓ | Time ↓ |
|---|---|---|---|---|
| 1 ($L = 0$) | 15.22 | 0.46 | 0.43 | **6 mins** |
| 2 ($L = 1$) | 16.58 | 0.53 | 0.37 | 7 mins |
| 3 ($L = 2$) | 18.07 | **0.54** | **0.35** | 10 mins |
| 4 ($L = 3$) | **18.08** | **0.54** | 0.36 | 15 mins |

Table 4: **Ablation of different components on the LLFF dataset with two input views.**

| Weight-sharing | $\mathcal{L}_{\text{ms-color}}$ | $\mathcal{L}_{\text{geo}}$ | $\mathbf{r}_{\text{novel}}$ | PSNR ↑ | SSIM ↑ | LPIPS ↓ | Model size ↓ |
|---|---|---|---|---|---|---|---|
| - | ✓ | ✓ | ✓ | 17.54 | 0.52 | 0.37 | 198.31 MB |
| ✓ | - | ✓ | ✓ | 16.89 | 0.44 | 0.46 | **183.04 MB** |
| ✓ | ✓ | - | ✓ | 15.97 | 0.49 | 0.41 | **183.04 MB** |
| ✓ | ✓ | ✓ | - | 17.84 | 0.52 | 0.36 | **183.04 MB** |
| ✓ | ✓ | ✓ | ✓ | **18.07** | **0.54** | **0.35** | **183.04 MB** |

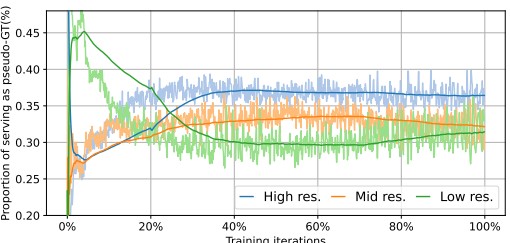 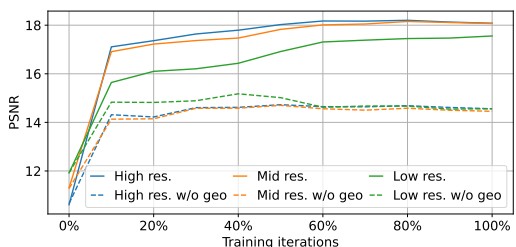

Figure 6: **Cross-scale geometric adaptation during training.** (*Left*) Low-resolution voxels initially guide geometry learning, with higher resolutions contributing more over time. This enables autonomous frequency tuning and better generalization. (*Right*) Geometric adaptation improves convergence quality across all scales compared to training without it.

SparseNeRF (Wang et al., 2023), ZeroRF[4] (Shi et al., 2024), and FSGS (Zhu et al., 2023) on the dataset preprocessed by pixelNeRF (Yu et al., 2021). Tab. 2 shows FrugalNeRF achieves state-of-the-art performance in most cases, with the shortest training time. Qualitative comparisons (Fig. 5) demonstrate FrugalNeRF's superior visual results, consistently rendering fine details (e.g., the blue elf's eyes) without noticeable artifacts, unlike other methods. This showcases FrugalNeRF's ability to model scenes with simple backgrounds effectively.

## 4.2 ABLATION STUDIES

**Number of scales.** We examine the effect of different numbers of scales in Tab. 3. The results show that by increasing the number of scales, we achieve better rendering quality. As there are more different resolutions of voxels, FrugalNeRF is more capable of representing different levels of details in the scene by geometric adaptation. We use $L = 2$ in our experiments, which indicates three scales in total, to strike a balance between rendering quality and training time.

**Weight-sharing voxels.** We compared the performance and memory usage of weight-sharing voxels against three independent voxels. Tab. 4 indicates that weight-sharing not only enhances performance but also reduces the model size.

**Multi-scale voxel color loss.** We demonstrate the effectiveness of multi-scale voxel color loss $\mathcal{L}_{\text{ms-color}}$ by comparing it to using color loss only on the largest scale (Tab. 4, Fig. 9(*Left*)). Multi-scale loss improves rendering and geometry by capturing various levels of scene detail. Without geometric adaptation, FrugalNeRF underperforms FreeNeRF, which uses a scheduling mechanism for gradually increasing input frequency. Our voxel grid representation offers faster training than MLPs but sacrifices some continuity. The discrete nature of multi-scale voxel grids initially limits our quality compared to FreeNeRF. However, integrating geometric adaptation significantly enhances coherence across scales, effectively overcoming this limitation.

**Cross-scale geometric adaptation.** Tab. 4 shows that the performance drops on all metrics without geometric adaptation loss $\mathcal{L}_{\text{geo}}$. Fig. 9 (*Mid*) demonstrate that geometric adaptation greatly suppresses floaters. Fig. 6 (*Left*) shows that during the training, low-frequency components from the low-resolution voxels first guide the coarse geometry. Then, mid-frequency and high-frequency components gradually increase their proportion of serving as pseudo-ground truth. Therefore, our FrugalNeRF could generalize better to diverse scenes without complex training scheduling. Fig. 6 (*Right*) further demonstrates that geometric adaptation helps all scales converge at superior qualities.

---

[4]The official ZeroRF implementation samples rays that lie in object masks during training. We remove this masked sampling for fair comparisons with other methods.

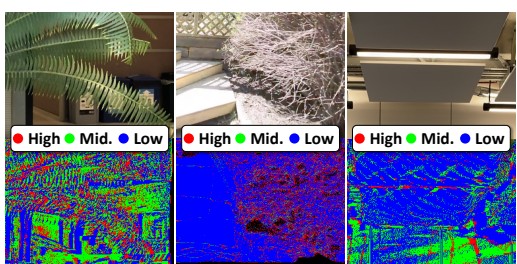 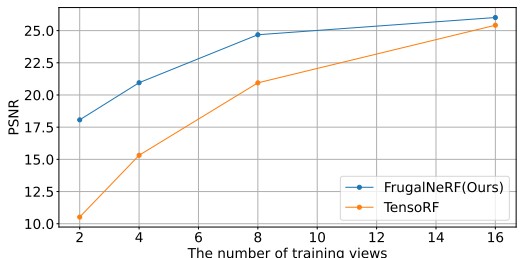

Figure 7: **Scene dependency analysis of multi-scale voxels.** ross-scale geometric adaptation can adapt to diverse scene configurations.

Figure 8: **Number of training views analysis.** FrugalNeRF significantly outperforms the base TensoRF representation on sparse views.

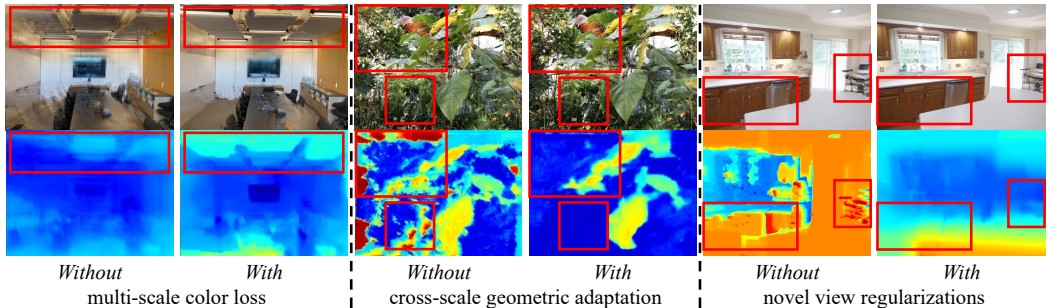

*Without* | *With*
multi-scale color loss

*Without* | *With*
cross-scale geometric adaptation

*Without* | *With*
novel view regularizations

Figure 9: **Visual comparisons on ablation studies.** (*Left*) Multi-scale color loss prevents overfitting and leads to a better result. (*Mid*) Geometric adaptation determines proper depth across scales via projection error and results in better geometry. (*Right*) Novel view regularizations provide additional supervisory signals from novel views and provide high-fidelity geometry.

**Scene dependency analysis of the multi-scale voxels.** We analyze the scene dependency of the multi-scale voxels in Fig. 7. The results indicate that scenes with foliage exhibit higher activations in high- and mid-frequency voxels, while textureless scenes show significant activations in low-frequency voxels. This confirms our approach's adaptability to different scene configurations.

**Number of training views analysis.** We plot the number of training views experiment in Fig. 8, demonstrating that FrugalNeRF outperforms TensoRF on sparse views (2 to 8 views) and continues to lead as the number of views increases.

**Novel view regularizations.** We evaluated the impact of novel view regularizations by omitting sample rays from novel views $r_{novel}$. Tab. 4 shows that using novel view rays and regularizations improves rendering quality. Fig. 9 (*Right*) illustrates that without these regularizations, training may get stuck in local minima, resulting in incorrect geometry. Novel view regularizations provide additional guidance, preventing overfitting and improving geometry accuracy.

## 5 CONCLUSION

In this paper, we propose FrugalNeRF, a framework that synthesizes novel views with extremely few input views. To speed up and regularize the training, we propose weight-sharing voxel representation across different scales, representing varying frequencies in the scene. To prevent overfitting, we propose a geometric adaptation scheme, utilizing reprojection errors to guide the geometry across different scales both in training and sampled novel views. FrugalNeRF performs on par with existing state-of-the-art methods on multiple datasets with shorter training time and does not rely on any externally learned priors.

**Limitations.** Few-shot NeRF relies on accurate camera poses for training. In scenarios with significant changes in viewpoint or sparse training views, the model may face challenges in generalization. Although our method introduces novel-view losses to deal with those unseen regions in training views, it is still an issue for few-shot NeRF.

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

# A  APPENDIX

## A.1  OVERVIEW

This supplementary material presents additional results to complement the main manuscript. First, we discuss the difference between competing methods in Appendix A.2 Second, we explain the implementation details in calculating reprojection errors in Appendix A.3. Then, we describe the details of adding pretrained monocular depth prior in Appendix A.4 Next, we provide all the training losses in our training process in Appendix A.5. Moreover, we describe the experimental setup, including the dataset and training time measurement of compared methods in our evaluations in Appendix A.6. In addition to this document, we provide an interactive HTML interface to compare our video results with state-of-the-art methods and show ablation videos and failure cases. We also attach the source code of our implementation for reference and will make it publicly available for reproducibility.

## A.2 DISCUSSIONS ON COMPETING MODELS

**GeCoNeRF.** GeCoNeRF (Kwak et al., 2023) is a few-shot NeRF that uses warped features as pseudo labels, which is sufficiently different from our method. Our method primarily focuses on cross-scale geometric adaptation, selecting render depths with minimal reprojection error across different scales as pseudo labels to adaptively learn the most suitable geometry for each scale. In contrast, GeCoNeRF, besides requiring a pre-trained feature extractor, directly optimizes warped features, making it highly sensitive to geometric noise and resulting in many floaters in its rendering result as shown in our supplementary videos. Our approach, on the other hand, is more robust due to our proposed multi-scale voxels. Low-resolution voxels represent coarse geometry, which is less likely to produce floaters. Using this as supervision effectively suppresses the generation of floaters.

**ZeroRF.** ZeroRF (Shi et al., 2024) is a concurrent work to ours, also aimed at training NeRF with sparse input views and achieving fast training times. Unlike TensoRF (Chen et al., 2022a), which directly optimizes the decomposed feature grid, ZeroRF parameterizes the feature grids with a randomly initialized deep neural network (generator). This decision is based on the belief in the higher resilience to noise and artifacts ability of deep neural networks. Although ZeroRF claims to achieve fast convergence stemming from its voxel representation, the need to train the generator results in slower training speeds compared to ours (refer to the main paper Table 2). Our method directly optimizes the feature grid and utilizes cross-scale geometry adaptation to avoid overfitting under sparse views, without requiring a generator that slows down convergence to form decomposed tensorial feature volumes. Additionally, we found that ZeroRF is not suitable for scenes with a background (*e.g.*, LLFF (Mildenhall et al., 2019a)) or datasets like the DTU (Jensen et al., 2014) Dataset, where ZeroRF must extensively use object masks for training. These object masks are not provided directly in these two datasets. Otherwise, ZeroRF may produce many artifacts and floaters, or the feature volume may be filled up to fit the background, leading to severe memory consumption issues causing training failures due to out-of-memory errors.

**SparseNeRF.** SparseNeRF (Wang et al., 2023) proposes a spatial continuity regularization that distills depth continuity priors, but it requires a pre-trained depth prior and is extremely slow by using MLP representation. Additionally, because monocular depth prediction results lack detail, SparseNeRF's rendered results tend to be blurry and lack detail. In contrast, our proposed cross-scale geometric adaptation does not rely on pre-trained priors and ensures the generation of overall geometry while paying attention to details.

**SimpleNeRF.** SimpleNeRF (Somraj et al., 2023) introduces a data augmentation method for few-shot NeRF, employing an MLP with fewer positional encoding frequencies for augmentation, but this simultaneously increases the training time. In contrast, we propose an efficient cross-scale geometric adaptation that achieves multi-scale representation through shared-weight voxels, eliminating the need for an additional model to reconstruct the same scene. This approach yields better results with lower costs.

**FreeNeRF.** FreeNeRF (Yang et al., 2023) is an MLP-based few-shot NeRF model. FreeNeRF proposes using a scheduling mechanism to gradually increase input frequency, allowing the model to learn low-frequency geometry during the early stages of training and then ramp up positional encoding to enable the model to learn more detailed geometry later on. However, our approach takes advantage of the explicit voxel representation, which converges faster and allows for direct cross-scaled geometry operations. Additionally, because we employ cross-scale geometry adaptation, our model dynamically determines which frequency of geometry to learn at different training stages. We do not require the complex frequency scheduling of FreeNeRF, nor are we limited to learning only high-frequency components in the later stages of training like FreeNeRF. This makes our method simpler, more general, and more robust.

**VGOS.** VGOS (Sun et al., 2023) introduces an incremental voxel training strategy and a voxel smoothing method for Few-shot NeRF, aimed at reducing training time. It employs a complex scheduling strategy to freeze the outer part of the voxel, leading to a leaky reconstruction of the background scene. Additionally, VGOS requires ground truth poses for novel pose sampling, which results in a quality drop when using random sampling. However, while VGOS's training time is

shorter than ours, its performance significantly lags behind. Our cross-scale geometric adaptation strategy eliminates the need for complex scheduling and ground truth pose sampling.

**FSGS.** FSGS (Zhu et al., 2023) addresses the challenge of limited 3D Gaussian splatting (3DGS) by introducing Proximity-guided Gaussian Unpooling, which adaptively densifies the Gaussians between existing points. Although this method mitigates the issue of insufficient GS, it still relies on a sufficient initial set of Gaussians to perform effectively. In few-shot scenarios, the initial number of GS can be extremely sparse, leading to suboptimal results. Furthermore, FSGS frequently requires novel view inference using monocular depth models during training, which significantly increases the training time. In contrast, our cross-scale geometric adaptation approach ensures rapid convergence without relying on novel view inference or monocular depth models, providing efficient and robust performance even with minimal initial data.

### A.3 DETAILS OF CALCULATING REPROJECTION ERRORS

Mathematically, let $\mathbf{p}_i$ be a 2D pixel coordinate in frame $i$, and $\widetilde{\mathbf{p}_i}$ be its homogeneous augmentation. The depth $D_i^l(\mathbf{p}_i)$ at scale $l$ obtained from volume rendering, and camera intrinsics $K_i$ are used to reproject $\mathbf{p}_i$ onto the 3D point $\mathbf{x}_i^l$ in camera coordinate system of frame $i$. Subsequently, utilizing the rotation matrix $R_i$ and translation matrix $t_i$ of frame $i$, $\mathbf{x}_i^l$ are transformed into world coordinates system $\mathbf{x}^l$:

$$\mathbf{x}_i^l = D_i^l(\mathbf{p}_i)K_i^{-1}\widetilde{\mathbf{p}}_i \tag{11}$$

$$\mathbf{x}^l = R_i\mathbf{x}_i^l + t_i \tag{12}$$

We simplify the previous two equations because the position of the 3D point $\mathbf{x}^l$ in world coordinates can also be determined directly from the ray defined by the starting point $o_i(\mathbf{p}_i)$ and the direction $v_i(\mathbf{p}_i)$:

$$\mathbf{x}^l = o_i(\mathbf{p}_i) + D_i^l(\mathbf{p}_i)v_i(\mathbf{p}_i) \tag{13}$$

Following this, the 3D point $\mathbf{x}^l$ in the world coordinate system is transformed to the camera coordinate system of frame $j$ using its rotation matrices $R_j$, and translation matrices $T_j$:

$$\mathbf{x}_{i\to j}^l = R_j^T\left(\mathbf{x}^l - t_j\right) \tag{14}$$

Finally, project it back to the 2D pixel coordinate system of frame $j$,

$$\widetilde{\mathbf{p}}_{i\to j}^l = \pi(K_j\mathbf{x}_{i\to j}^l) \tag{15}$$

where $\pi([x, y, z]^T) = \left[\frac{x}{z}, \frac{y}{z}\right]$. Using coordinates $\mathbf{p}_i$ and $\mathbf{p}_{i\to j}^l$ to index the RGB maps of frames $i$ (denoted as $C_i$) and $j$ (denoted as $C_j$), facilitating the computation of the reprojection error:

$$e^l(\mathbf{p}_i) = \left\|C_i(\mathbf{p}_i) - C_j(\mathbf{p}_{i\to j}^l)\right\|^2 \tag{16}$$

Therefore, for each ray sampled from the training view, the pseudo-GT depth of the scale with the minimum reprojection error is obtained,

$$D'(\mathbf{r}_{\text{train}}) = \arg\min_l(e^l(\mathbf{r}_{\text{train}})). \tag{17}$$

where the pseudo-GT depth is utilized to compute the geometric adaptation loss (MSE) $\mathcal{L}_{\text{geo}}$.

$$\mathcal{L}_{\text{geo}}(\mathbf{r}_{\text{train}}) = \sum_{l=0}^{L}\sum_{r_{\text{train}}\in\mathcal{R}_{\text{train}}}\left\|\hat{D}^l(\mathbf{r}_{\text{train}}) - D'(\mathbf{r}_{\text{train}})\right\|^2. \tag{18}$$

This mechanism provides a supervisory signal for geometry, ensuring that the model can effectively maintain the geometric integrity of the scene across different scales, even in the absence of explicit depth ground truth. It is a pivotal part of the training process, allowing the model to adapt and refine its understanding of the scene's geometric structure in a self-adaptive manner. In our implementation, instead of using a single pixel to calculate reprojection error, we use a patch with $5 \times 5$ pixels to calculate reprojection error. This avoids warping noise caused by similar patterns in scenes, for example, in the case of the LLFF fortress and room. Furthermore, we set a threshold for reprojection error that allows us to ignore cases of image warping with occlusions and prevents crashes during initial training processes, which typically have high reprojection errors.

### A.4 Details of adding Pretrained Monocular Depth Prior

We utilize the pre-trained Dense Prediction Transformer (DPT) (Ranftl et al., 2021b) to generate monocular depth maps from training views. DPT is trained on 1.4 million image-depth pairs, making it a convenient and effective choice for our setup. To address the scale ambiguity between the true scene scale and the estimated depth, we introduce a relaxed relative loss based on Pearson correlation between the estimated and rendered depth maps. This loss is applied at multiple scales, enhancing the monocular depth prior's constraint across different scales and improving the overall geometric consistency.

### A.5 Losses

**Voxel TV loss ($\mathcal{L}_{\mathbf{tv}}$).** We use the TV loss on voxel to smooth the result in voxel space.

**Patch-wise depth smoothness loss ($\mathcal{L}_{\mathbf{ds}}$).** We sample patches of rays and calculate the total variance of depth to smooth the geometry in the depth space.

**L1 sparsity loss ($\mathcal{L}_{\mathbf{l1}}$).** We suppress the voxel density in air space by introducing a density L1 regularization loss.

**Distortion loss ($\mathcal{L}_{\mathbf{dist}}$).** We adopt the approach from Mip-NeRF 360 (Barron et al., 2022), integrating distortion loss to remove floaters from the novel views.

**Occlusion loss ($\mathcal{L}_{\mathbf{occ}}$).** In the DTU dataset, we follow FreeNeRF Yang et al. (2023) by incorporating an occlusion loss that utilizes black and white background priors to push floaters into the background.

**Novel pose sampling form spiraling trajectory.** We follow the implementation of a spiraling trajectory from TensoRF (Chen et al., 2022a). For the LLFF dataset, we sample 60 novel poses from the spiraling trajectory sampled from training views with 1 rotations, radius scale 1.0, and $z_{\mathrm{rate}}$ 0.5. For the DTU dataset, we sample 60 novel poses from the spiraling trajectory sampled from training views with 4 rotations, radius scale 0.5, and $z_{\mathrm{rate}}$ 0.5. For the RealEstate-10K dataset, we sample 60 novel poses from the spiraling trajectory sampled from training views with 2 rotations, radius scale 2.0, and $z_{\mathrm{rate}}$ 0.5.

### A.6 Experimental Setup

We compare the result of few-shot NeRF on LLFF and DTU with $n = 2, 3, 4$ input views.

**LLFF dataset.** The LLFF dataset comprises 8 forward-facing unbounded scenes with variable frame counts at a resolution of $1008 \times 756$. In line with prior work (Somraj & Soundararajan, 2023), we use every 8th frame for testing in each scene. For training, we uniformly sample $n$ views from the remaining frames.

**DTU dataset.** The DTU dataset is a large-scale multi-view collection that includes 124 different scenes. Follow the Pixel-NeRF (Yu et al., 2021) and ViP-NeRF (Somraj & Soundararajan, 2023) approach, we use the same test sets. However, because COLMAP will fail to generate sparse depth at scans 8, 30, and 110, we can only test on 12 scenes. Test scan IDs are 21, 31, 34, 38, 40, 41, 45, 55, 63, 82, 103, and 114. We use specific image IDs as input views and downsample images to $300 \times 400$ pixels for consistency with prior studies (Yu et al., 2021; Somraj & Soundararajan, 2023).

**RealEstate-10K dataset.** RealEstate-10K is a comprehensive database of approximately 80,000 video segments, each with over 30 frames, widely utilized for novel view synthesis. For our study, we select five scenes from its extensive test set, following the approach outlined in ViP-NeRF (Somraj & Soundararajan, 2023). We selected frames 0, 10, 20, and 30 for the training set with a resolution of $1024 \times 576$, in accordance with the SimpleNeRF (Somraj et al., 2023) methodology, while testing on the same test set as SimpleNeRF (Somraj et al., 2023) due to the unobserved region problem, which NeRF cannot handle, in some testing view.

### A.6.1 Training Time Measurement and Time Complexity

**RegNeRF.**   We use the official implementation of RegNeRF (Niemeyer et al., 2022) and follow most of the default configuration, while the batch size or other hyperparameters might be adjusted due to the GPU memory issue. For the LLFF dataset, the training requires roughly 2.35 hours per scene with 69769 iterations and a batch size of 2,048. Note that RegNeRF samples 10000 random poses by its default configuration on the DTU dataset, leading to out-of-memory on a single NVIDIA RTX 4090 GPU. While reducing the number of random poses to about 1/8 could potentially resolve this issue, such a reduction is likely to adversely affect the performance, so we simply exclude this method from our experiments.

**FreeNeRF.**   We use the official implementation of FreeNeRF (Yang et al., 2023) and follow most of the default configuration, while the batch size or other hyperparameters might be adjusted due to the GPU memory issue. For the LLFF dataset, the training requires roughly 1.5 hours per scene with 69,769 iterations and a batch size of 2,048. For the DTU dataset, the training requires about 1 hour per scene with 43,945 iterations and a batch size of 2,048.

**SparseNeRF.**   We use the official implementation of SparseNeRF. (Wang et al., 2023) and follow most of the default configuration, while the batch size or other hyperparameters might be adjusted due to the GPU memory issue. For the LLFF dataset, the training requires roughly 1 hour per scene with 70,000 iterations and a batch size of 512. For the DTU dataset, the training requires about 30 minutes per scene with 70,000 iterations and a batch size of 256.

**SimpleNeRF.**   We use the official implementation of SimpleNeRF (Somraj et al., 2023) and follow most of the default configuration, while the batch size or other hyperparameters might be adjusted due to the GPU memory issue. For the LLFF dataset, we use the model weights released by the author directly. Since there's no official implemented dataloader for the DTU dataset, we use the dataloader and configuration from ViP-NeRF (Somraj & Soundararajan, 2023), which requires about 1.38 hours per scene with 25,000 iterations and batch size of 2,048.

**VGOS.**   We furter provide VGOS result. We use the official implementation of VGOS (Sun et al., 2023) and follow most of the default configuration, while the batch size or other hyperparameters might be adjusted due to the GPU memory issue. Note that VGOS samples random poses directly from the entire dataset, which is unreasonable under the few-shot setting, so we replace the sampling with the interpolation from training poses implemented in the official repo. For the LLFF dataset, the training requires roughly 5 minutes per scene with 9,000 iterations and a batch size of 16,384. For the DTU dataset, the training requires about 3 minutes per scene with 9,000 iterations and a batch size of 16,384. Note that VGOS seems invalid on the DTU dataset (Fig. 11) and they does not evaluate the DTU dataset in their paper.

**GeCoNeRF.**   As mentioned in GeCoNeRF (Kwak et al., 2023)'s official github repo, their current code is unexecutable. To complete our experiment, we still try our best to implement their method based on the code provided. For the LLFF dataset, the training requires roughly 4 hours per scene with 85,000 iterations and a batch size of 1024. It is important to note that we utilized 2 GPUs for training this method, so the training time reported in our paper might be shorter than what is actually required.

**ZeroRF.**   We use the official implementation of ZeroRF (Shi et al., 2024) and follow most of the default configurations. For the LLFF dataset, ZeroRF does not provide the dataloader for the LLFF, and their paper mentions its inability to be used for unbounded scenes. Therefore, our primary testing was conducted on the DTU dataset. In the DTU dataset, the original implementation of ZeroRF necessitates masking out the background area of the input frame before training, which is incompatible with our evaluation benchmark. Consequently, we trained it without object masks. Training requires approximately 25 minutes per scene with 10,000 iterations and a batch size of $2^{14}$.

**FSGS.**   We use the official implementation of FSGS (Zhu et al., 2023) and follow most of the default configurations. For the LLFF dataset, we adjust the input views to match the settings used in ViP-NeRF, which differs from the original FSGS paper. Training takes approximately 25 minutes per

Table 5: **Comparison of the time complexity.**

| Method | MFLOPs / pixel ↓ |
|---|---|
| FreeNeRF (Yang et al., 2023) | 288.57 |
| ViP-NeRF (Somraj & Soundararajan, 2023) | 149.26 |
| SimpleNeRF (Somraj et al., 2023) | 303.82 |
| SparseNeRF (Wang et al., 2023) | 287.92 |
| Ours | 13.77 |

Table 6: **Quantitative results on the LLFF (Mildenhall et al., 2019a) dataset with two input views. The three rows show LPIPS, SSIM, and PSNR scores, respectively.**

| Method \ Scene | Fern | Flower | Fortress | Horns | Leaves | Orchids | Room | Trex | Average |
|---|---|---|---|---|---|---|---|---|---|
| RegNeRF (Niemeyer et al., 2022) | 0.51 | 0.43 | 0.37 | 0.51 | 0.35 | 0.45 | 0.38 | 0.42 | 0.43 |
| | 0.45 | 0.51 | 0.46 | 0.42 | 0.37 | 0.30 | 0.74 | 0.54 | 0.49 |
| | 15.8 | 17.0 | 20.6 | 15.9 | 14.5 | 13.9 | 18.7 | 16.7 | 16.9 |
| DS-NeRF (Deng et al., 2022) | 0.50 | 0.43 | 0.30 | 0.49 | 0.47 | 0.43 | 0.35 | 0.41 | 0.42 |
| | 0.46 | 0.44 | 0.65 | 0.49 | 0.24 | 0.32 | 0.76 | 0.53 | 0.51 |
| | 16.4 | 16.1 | 23.0 | 16.6 | 12.4 | 13.7 | 18.9 | 15.7 | 16.9 |
| DDP-NeRF (Roessle et al., 2022) | 0.44 | 0.46 | 0.17 | 0.46 | 0.52 | 0.41 | 0.30 | 0.43 | 0.39 |
| | 0.49 | 0.45 | 0.77 | 0.52 | 0.23 | 0.38 | 0.76 | 0.54 | 0.54 |
| | 17.2 | 16.2 | 22.7 | 17.1 | 12.6 | 15.1 | 18.7 | 15.7 | 17.2 |
| FreeNeRF (Yang et al., 2023) | 0.46 | 0.38 | 0.33 | 0.43 | 0.36 | 0.42 | 0.34 | 0.33 | 0.38 |
| | 0.49 | 0.55 | 0.53 | 0.53 | 0.38 | 0.35 | 0.76 | 0.60 | 0.54 |
| | 17.1 | 17.6 | 21.3 | 17.1 | 14.4 | 14.1 | 18.3 | 18.1 | 17.6 |
| ViP-NeRF (Somraj & Soundararajan, 2023) | 0.45 | 0.42 | 0.21 | 0.39 | 0.46 | 0.40 | 0.36 | 0.38 | 0.37 |
| | 0.45 | 0.43 | 0.71 | 0.54 | 0.21 | 0.36 | 0.72 | 0.54 | 0.52 |
| | 16.2 | 14.9 | 22.6 | 17.1 | 11.7 | 14.2 | 17.7 | 15.9 | 16.7 |
| SimpleNeRF (Somraj et al., 2023) | 0.51 | 0.43 | 0.25 | 0.42 | 0.44 | 0.41 | 0.35 | 0.39 | 0.39 |
| | 0.50 | 0.53 | 0.67 | 0.54 | 0.30 | 0.37 | 0.77 | 0.58 | 0.55 |
| | 17.0 | 16.9 | 22.5 | 17.1 | 13.5 | 14.7 | 19.5 | 16.8 | 17.6 |
| VGOS (Sun et al., 2023) | 0.48 | 0.44 | 0.37 | 0.47 | 0.36 | 0.42 | 0.38 | 0.40 | 0.42 |
| | 0.51 | 0.55 | 0.53 | 0.55 | 0.38 | 0.40 | 0.77 | 0.59 | 0.55 |
| | 16.5 | 17.5 | 19.4 | 15.7 | 14.7 | 14.4 | 18.8 | 16.0 | 16.7 |
| GeCoNeRF (Kwak et al., 2023) | 0.56 | 0.49 | 0.50 | 0.61 | 0.49 | 0.51 | 0.54 | 0.49 | 0.52 |
| | 0.47 | 0.49 | 0.43 | 0.41 | 0.28 | 0.29 | 0.68 | 0.52 | 0.45 |
| | 16.4 | 16.9 | 17.9 | 15.4 | 13.3 | 13.4 | 17.3 | 16.1 | 15.8 |
| SparseNeRF (Wang et al., 2023) | 0.48 | 0.55 | 0.40 | 0.52 | 0.52 | 0.55 | 0.29 | 0.37 | 0.45 |
| | 0.52 | 0.41 | 0.61 | 0.51 | 0.244 | 0.24 | 0.82 | 0.62 | 0.52 |
| | 18.2 | 15.4 | 21.7 | 17.4 | 13.4 | 13.3 | 22.8 | 18.6 | 18.0 |
| FSGS (Zhu et al., 2023) | 0.46 | 0.45 | 0.35 | 0.42 | 0.33 | 0.41 | 0.38 | 0.45 | 0.41 |
| | 0.40 | 0.38 | 0.47 | 0.42 | 0.34 | 0.24 | 0.72 | 0.46 | 0.45 |
| | 15.0 | 14.8 | 16.9 | 16.2 | 14.2 | 12.6 | 17.6 | 13.8 | 15.3 |
| FrugalNeRF (Ours) | 0.41 | 0.41 | 0.27 | 0.36 | 0.32 | 0.42 | 0.34 | 0.32 | 0.35 |
| | 0.47 | 0.50 | 0.54 | 0.55 | 0.41 | 0.33 | 0.75 | 0.61 | 0.54 |
| | 17.4 | 17.5 | 20.3 | 18.5 | 15.5 | 15.0 | 19.2 | 18.6 | 18.1 |
| FrugalNeRF w/ mono. depth (Ours) | 0.40 | 0.40 | 0.27 | 0.37 | 0.33 | 0.39 | 0.32 | 0.35 | 0.35 |
| | 0.46 | 0.53 | 0.54 | 0.54 | 0.41 | 0.37 | 0.76 | 0.59 | 0.54 |
| | 17.7 | 17.9 | 20.9 | 18.5 | 15.4 | 15.6 | 19.6 | 18.2 | 18.3 |

scene with 10,000 iterations. Since there is no official dataloader for the DTU dataset, we convert the DTU camera poses to the LLFF format and use the default LLFF configuration. Training on the DTU dataset requires around 20 minutes per scene with 10,000 iterations.

**Time complexity.** To verify the efficiency of our method, besides comparing the training time of various methods, we also calculated the MFLOPs per pixel in Tab. 5.

A.7 COMPLETE QUANTITATIVE EVALUATIONS

**LLFF dataset.** We show all 8 scenes of the quantitative comparisons with two, three, and four input views on the LLFF dataset in Tab. 6, Tab. 7, and Tab. 8, respectively.

**DTU dataset.** We show all 12 scenes of the quantitative comparisons with two, three, and four input views on the LLFF dataset in Tab. 9, Tab. 10, and Tab. 11, respectively.

Table 7: **Quantitative results on the LLFF (Mildenhall et al., 2019a) dataset with three input views. The three rows show LPIPS, SSIM, and PSNR scores, respectively.**

| Method | Scene | Fern | Flower | Fortress | Horns | Leaves | Orchids | Room | Trex | Average |
|---|---|---|---|---|---|---|---|---|---|---|
| RegNeRF (Niemeyer et al., 2022) | | 0.47 | 0.27 | 0.31 | 0.44 | 0.39 | 0.44 | 0.25 | 0.36 | 0.36 |
| | | 0.48 | 0.58 | 0.64 | 0.53 | 0.37 | 0.31 | 0.81 | 0.63 | 0.57 |
| | | 17.9 | 19.6 | 22.7 | 18.2 | 14.6 | 14.2 | 21.0 | 18.4 | 18.7 |
| DS-NeRF (Deng et al., 2022) | | 0.47 | 0.25 | 0.25 | 0.47 | 0.50 | 0.45 | 0.22 | 0.37 | 0.36 |
| | | 0.52 | 0.66 | 0.72 | 0.52 | 0.25 | 0.33 | 0.84 | 0.59 | 0.58 |
| | | 18.5 | 21.3 | 24.8 | 17.5 | 12.6 | 14.1 | 23.0 | 17.1 | 19.0 |
| DDP-NeRF (Roessle et al., 2022) | | 0.47 | 0.29 | 0.20 | 0.48 | 0.52 | 0.45 | 0.32 | 0.42 | 0.39 |
| | | 0.53 | 0.63 | 0.75 | 0.53 | 0.24 | 0.35 | 0.76 | 0.54 | 0.56 |
| | | 18.5 | 20.2 | 22.1 | 17.4 | 12.8 | 15.1 | 18.3 | 16.0 | 17.7 |
| FreeNeRF (Yang et al., 2023) | | 0.40 | 0.28 | 0.32 | 0.41 | 0.40 | 0.41 | 0.22 | 0.33 | 0.34 |
| | | 0.54 | 0.61 | 0.60 | 0.58 | 0.40 | 0.37 | 0.85 | 0.64 | 0.60 |
| | | 18.9 | 20.7 | 22.0 | 18.7 | 15.0 | 14.7 | 22.6 | 19.0 | 19.3 |
| ViP-NeRF (Somraj & Soundararajan, 2023) | | 0.51 | 0.24 | 0.19 | 0.42 | 0.44 | 0.41 | 0.27 | 0.32 | 0.34 |
| | | 0.49 | 0.65 | 0.76 | 0.57 | 0.25 | 0.34 | 0.81 | 0.62 | 0.59 |
| | | 17.3 | 20.8 | 24.5 | 18.2 | 12.4 | 14.2 | 21.7 | 18.1 | 18.9 |
| SimpleNeRF (Somraj et al., 2023) | | 0.43 | 0.24 | 0.17 | 0.42 | 0.42 | 0.39 | 0.26 | 0.34 | 0.33 |
| | | 0.52 | 0.66 | 0.78 | 0.57 | 0.38 | 0.38 | 0.83 | 0.66 | 0.62 |
| | | 18.2 | 20.7 | 24.7 | 18.4 | 14.8 | 15.0 | 22.0 | 18.9 | 19.5 |
| VGOS (Sun et al., 2023) | | 0.40 | 0.31 | 0.33 | 0.46 | 0.40 | 0.41 | 0.31 | 0.35 | 0.37 |
| | | 0.58 | 0.61 | 0.69 | 0.58 | 0.40 | 0.40 | 0.83 | 0.66 | 0.61 |
| | | 19.0 | 20.0 | 23.0 | 17.0 | 15.0 | 15.2 | 21.8 | 18.0 | 18.8 |
| GeCoNeRF (Kwak et al., 2023) | | 0.57 | 0.36 | 0.45 | 0.60 | 0.50 | 0.51 | 0.34 | 0.43 | 0.47 |
| | | 0.46 | 0.57 | 0.53 | 0.44 | 0.32 | 0.30 | 0.80 | 0.59 | 0.50 |
| | | 17.0 | 19.5 | 20.6 | 15.8 | 13.8 | 13.6 | 21.1 | 18.1 | 17.4 |
| SparseNeRF (Wang et al., 2023) | | 0.43 | 0.33 | 0.37 | 0.50 | 0.35 | 0.41 | 0.28 | 0.31 | 0.37 |
| | | 0.57 | 0.60 | 0.59 | 0.53 | 0.45 | 0.37 | 0.81 | 0.67 | 0.59 |
| | | 19.6 | 19.8 | 23.0 | 18.4 | 16.5 | 15.2 | 21.5 | 20.1 | 19.5 |
| FSGS (Zhu et al., 2023) | | 0.48 | 0.30 | 0.15 | 0.36 | 0.26 | 0.35 | 0.28 | 0.28 | 0.30 |
| | | 0.55 | 0.68 | 0.72 | 0.65 | 0.28 | 0.37 | 0.84 | 0.62 | 0.61 |
| | | 17.9 | 21.5 | 23.9 | 19.4 | 13.3 | 14.1 | 22.6 | 17.4 | 19.2 |
| FrugalNeRF (Ours) | | 0.39 | 0.32 | 0.24 | 0.34 | 0.37 | 0.42 | 0.27 | 0.29 | 0.32 |
| | | 0.50 | 0.55 | 0.63 | 0.59 | 0.39 | 0.35 | 0.81 | 0.66 | 0.59 |
| | | 18.2 | 18.8 | 23.4 | 19.3 | 15.5 | 15.3 | 22.2 | 19.3 | 19.4 |
| FrugalNeRF w/ mono. depth (Ours) | | 0.40 | 0.23 | 0.22 | 0.33 | 0.37 | 0.40 | 0.25 | 0.29 | 0.30 |
| | | 0.49 | 0.63 | 0.69 | 0.60 | 0.39 | 0.36 | 0.83 | 0.67 | 0.61 |
| | | 18.6 | 21.4 | 23.5 | 19.0 | 15.4 | 15.7 | 22.3 | 20.0 | 19.9 |

**RealEstate-10K dataset.** We show all 12 scenes of the quantitative comparisons with two, three, and four input views on the LLFF dataset in Tab. 12, Tab. 13, Tab. 14, and Tab. 15.

## A.8 ADDITIONAL VISUAL COMPARISONS

**LLFF dataset.** We show additional visual comparisons on the LLFF dataset with two input views in Fig. 10.

**DTU dataset.** We show additional visual comparisons on the DTU dataset with two input views in Fig. 11.

**RealEstate-10K dataset.** We further present the qualitative comparisons of novel view synthesis on the RealEstate-10K dataset with two input views in Fig. 13. Compared to SimpleNeRF (Somraj et al., 2023), which requires hours of training, FrugalNeRF needs only less than 20 minutes and can render comparable results, demonstrating FrugalNeRF's effectiveness in more in-the-wild scenes.

Table 8: **Quantitative results on the LLFF (Mildenhall et al., 2019a) dataset with four input views. The three rows show LPIPS, SSIM, and PSNR scores, respectively.**

| Method | Scene / Fern | Flower | Fortress | Horns | Leaves | Orchids | Room | Trex | Average |
|---|---|---|---|---|---|---|---|---|---|
| RegNeRF (Niemeyer et al., 2022) | 0.35 | 0.29 | 0.37 | 0.34 | 0.32 | 0.43 | 0.19 | 0.32 | 0.32 |
| | 0.63 | 0.64 | 0.55 | 0.64 | 0.44 | 0.34 | 0.87 | 0.66 | 0.62 |
| | 20.8 | 19.8 | 22.4 | 20.1 | 15.9 | 14.8 | 23.9 | 18.9 | 19.9 |
| DS-NeRF (Deng et al., 2022) | 0.35 | 0.28 | 0.31 | 0.41 | 0.41 | 0.41 | 0.16 | 0.39 | 0.34 |
| | 0.63 | 0.64 | 0.66 | 0.59 | 0.39 | 0.38 | 0.89 | 0.59 | 0.61 |
| | 20.9 | 20.6 | 24.1 | 19.5 | 15.8 | 15.2 | 25.6 | 17.1 | 20.1 |
| DDP-NeRF (Roessle et al., 2022) | 0.40 | 0.30 | 0.18 | 0.42 | 0.45 | 0.42 | 0.26 | 0.39 | 0.35 |
| | 0.60 | 0.63 | 0.73 | 0.59 | 0.37 | 0.41 | 0.82 | 0.60 | 0.61 |
| | 20.1 | 20.0 | 23.4 | 19.3 | 15.1 | 15.8 | 20.8 | 17.3 | 19.2 |
| FreeNeRF (Yang et al., 2023) | 0.37 | 0.30 | 0.35 | 0.37 | 0.35 | 0.42 | 0.19 | 0.31 | 0.33 |
| | 0.64 | 0.64 | 0.60 | 0.63 | 0.47 | 0.37 | 0.88 | 0.68 | 0.63 |
| | 21.1 | 20.5 | 23.2 | 20.4 | 16.6 | 14.9 | 24.8 | 19.6 | 20.5 |
| ViP-NeRF (Somraj & Soundararajan, 2023) | 0.39 | 0.27 | 0.25 | 0.38 | 0.36 | 0.40 | 0.23 | 0.32 | 0.32 |
| | 0.58 | 0.63 | 0.70 | 0.60 | 0.40 | 0.39 | 0.85 | 0.64 | 0.62 |
| | 18.2 | 19.5 | 23.3 | 19.0 | 14.8 | 14.8 | 23.2 | 18.6 | 19.3 |
| SimpleNeRF (Somraj et al., 2023) | 0.33 | 0.27 | 0.28 | 0.38 | 0.35 | 0.36 | 0.19 | 0.32 | 0.31 |
| | 0.65 | 0.67 | 0.69 | 0.63 | 0.46 | 0.42 | 0.88 | 0.68 | 0.65 |
| | 21.1 | 20.8 | 24.3 | 19.7 | 16.3 | 15.7 | 24.3 | 19.3 | 20.4 |
| VGOS (Sun et al., 2023) | 0.40 | 0.35 | 0.40 | 0.43 | 0.34 | 0.41 | 0.28 | 0.35 | 0.37 |
| | 0.64 | 0.63 | 0.64 | 0.62 | 0.49 | 0.43 | 0.86 | 0.68 | 0.64 |
| | 19.6 | 20.3 | 22.7 | 18.6 | 16.6 | 15.8 | 23.6 | 18.7 | 19.7 |
| GeCoNeRF (Kwak et al., 2023) | 0.45 | 0.36 | 0.44 | 0.47 | 0.44 | 0.51 | 0.27 | 0.40 | 0.42 |
| | 0.61 | 0.61 | 0.51 | 0.40 | 0.40 | 0.30 | 0.85 | 0.63 | 0.56 |
| | 20.5 | 19.9 | 21.2 | 19.6 | 15.5 | 13.9 | 23.5 | 19.0 | 19.1 |
| SparseNeRF (Wang et al., 2023) | 0.42 | 0.32 | 0.31 | 0.39 | 0.36 | 0.42 | 0.25 | 0.29 | 0.34 |
| | 0.62 | 0.64 | 0.70 | 0.63 | 0.49 | 0.39 | 0.85 | 0.70 | 0.65 |
| | 21.4 | 20.7 | 24.6 | 20.4 | 17.5 | 15.7 | 23.5 | 20.9 | 20.9 |
| FSGS (Zhu et al., 2023) | 0.26 | 0.22 | 0.17 | 0.24 | 0.22 | 0.28 | 0.17 | 0.23 | 0.22 |
| | 0.67 | 0.65 | 0.65 | 0.70 | 0.46 | 0.45 | 0.88 | 0.71 | 0.66 |
| | 20.5 | 20.2 | 22.6 | 20.9 | 15.6 | 15.4 | 23.7 | 19.2 | 20.1 |
| FrugalNeRF (Ours) | 0.30 | 0.28 | 0.24 | 0.30 | 0.26 | 0.38 | 0.19 | 0.27 | 0.27 |
| | 0.63 | 0.64 | 0.66 | 0.66 | 0.52 | 0.41 | 0.87 | 0.72 | 0.65 |
| | 21.1 | 20.8 | 23.6 | 21.6 | 16.9 | 16.3 | 24.2 | 19.7 | 20.9 |
| FrugalNeRF w/ mono. depth (Ours) | 0.30 | 0.27 | 0.25 | 0.28 | 0.24 | 0.37 | 0.18 | 0.27 | 0.26 |
| | 0.64 | 0.65 | 0.64 | 0.68 | 0.53 | 0.41 | 0.88 | 0.71 | 0.66 |
| | 21.5 | 20.9 | 23.9 | 21.1 | 17.2 | 16.3 | 24.1 | 19.6 | 20.9 |

Table 9: **Quantitative results on the DTU Jensen et al. (2014) dataset with two input views. The three rows show LPIPS, SSIM and PSNR scores, respectively.**

| Method | Scene / Scan21 | Scan31 | Scan34 | Scan38 | Scan40 | Scan41 | Scan45 | Scan55 | Scan63 | Scan82 | Scan103 | Scan114 | Average |
|---|---|---|---|---|---|---|---|---|---|---|---|---|---|
| FreeNeRF Yang et al. (2023) | 0.33 | 0.18 | 0.31 | 0.34 | 0.41 | 0.35 | 0.19 | 0.11 | 0.07 | 0.08 | 0.17 | 0.12 | 0.22 |
| | 0.51 | 0.75 | 0.63 | 0.61 | 0.58 | 0.63 | 0.76 | 0.80 | 0.93 | 0.90 | 0.82 | 0.85 | 0.73 |
| | 13.21 | 19.33 | 14.66 | 16.76 | 11.42 | 14.50 | 18.66 | 21.62 | 23.19 | 21.56 | 17.55 | 24.19 | 18.05 |
| ViP-NeRF Somraj & Soundararajan (2023) | 0.37 | 0.24 | 0.27 | 0.38 | 0.31 | 0.23 | 0.31 | 0.21 | 0.09 | 0.12 | 0.18 | 0.17 | 0.24 |
| | 0.26 | 0.49 | 0.52 | 0.43 | 0.47 | 0.58 | 0.37 | 0.39 | 0.63 | 0.57 | 0.65 | 0.49 | 0.49 |
| | 11.31 | 13.57 | 17.13 | 13.25 | 15.08 | 17.81 | 11.35 | 16.92 | 16.71 | 13.37 | 16.15 | 16.24 | 14.91 |
| SimpleNeRF Somraj et al. (2023) | 0.23 | 0.32 | 0.23 | 0.21 | 0.24 | 0.19 | 0.28 | 0.22 | 0.30 | 0.27 | 0.19 | 0.27 | 0.25 |
| | 0.73 | 0.71 | 0.76 | 0.77 | 0.77 | 0.84 | 0.70 | 0.88 | 0.75 | 0.79 | 0.81 | 0.82 | 0.79 |
| | 12.71 | 11.91 | 14.39 | 14.50 | 13.76 | 15.57 | 11.88 | 19.58 | 12.73 | 14.37 | 16.64 | 14.86 | 14.41 |
| VGOS Sun et al. (2023) | 0.28 | 0.36 | 0.33 | 0.31 | 0.30 | 0.27 | 0.37 | 0.15 | 0.49 | 0.45 | 0.34 | 0.18 | 0.32 |
| | 0.69 | 0.67 | 0.69 | 0.71 | 0.73 | 0.78 | 0.64 | 0.90 | 0.56 | 0.57 | 0.73 | 0.85 | 0.71 |
| | 9.69 | 8.97 | 9.75 | 10.27 | 8.79 | 9.75 | 7.54 | 19.24 | 5.17 | 5.63 | 11.29 | 15.81 | 10.16 |
| SparseNeRF Wang et al. (2023) | 0.39 | 0.22 | 0.26 | 0.33 | 0.24 | 0.21 | 0.20 | 0.14 | 0.08 | 0.08 | 0.15 | 0.13 | 0.20 |
| | 0.45 | 0.69 | 0.70 | 0.60 | 0.72 | 0.76 | 0.75 | 0.78 | 0.92 | 0.91 | 0.84 | 0.85 | 0.75 |
| | 14.25 | 17.95 | 20.65 | 17.93 | 16.33 | 20.13 | 18.22 | 22.29 | 20.70 | 23.46 | 21.70 | 24.40 | 19.83 |
| ZeroRF Shi et al. (2024) | 0.45 | 0.27 | 0.35 | 0.44 | 0.29 | 0.28 | 0.39 | 0.25 | 0.13 | 0.18 | 0.25 | 0.29 | 0.30 |
| | 0.30 | 0.61 | 0.50 | 0.39 | 0.59 | 0.63 | 0.49 | 0.68 | 0.88 | 0.82 | 0.73 | 0.63 | 0.60 |
| | 10.99 | 14.40 | 13.93 | 12.16 | 15.41 | 16.73 | 11.24 | 17.08 | 20.39 | 15.36 | 16.23 | 14.12 | 14.84 |
| FrugalNeRF (Ours) | 0.25 | 0.16 | 0.20 | 0.24 | 0.24 | 0.17 | 0.16 | 0.13 | 0.09 | 0.07 | 0.13 | 0.11 | 0.16 |
| | 0.57 | 0.73 | 0.73 | 0.64 | 0.73 | 0.78 | 0.77 | 0.86 | 0.92 | 0.92 | 0.85 | 0.89 | 0.78 |
| | 14.67 | 17.86 | 19.47 | 17.66 | 14.51 | 19.74 | 16.94 | 24.87 | 21.21 | 22.67 | 21.45 | 25.60 | 19.72 |
| FrugalNeRF w/ mono. depth (Ours) | 0.25 | 0.15 | 0.19 | 0.21 | 0.23 | 0.16 | 0.15 | 0.12 | 0.08 | 0.07 | 0.10 | 0.10 | 0.15 |
| | 0.56 | 0.73 | 0.75 | 0.68 | 0.74 | 0.79 | 0.78 | 0.86 | 0.93 | 0.91 | 0.88 | 0.90 | 0.79 |
| | 14.14 | 18.46 | 21.27 | 19.40 | 15.56 | 20.53 | 18.05 | 25.65 | 23.46 | 22.72 | 23.76 | 26.25 | 20.77 |

Table 10: **Quantitative results on the DTU (Jensen et al., 2014) dataset with three input views. The three rows show LPIPS, SSIM and PSNR scores, respectively.**

| Method \ Scene | Scan21 | Scan31 | Scan34 | Scan38 | Scan40 | Scan41 | Scan45 | Scan55 | Scan63 | Scan82 | Scan103 | Scan114 | Average |
|---|---|---|---|---|---|---|---|---|---|---|---|---|---|
| FreeNeRF (Yang et al., 2023) | 15.93 | 19.53 | 23.23 | 19.88 | 18.38 | 22.83 | 21.07 | 22.88 | 25.28 | 26.39 | 26.68 | 26.68 | 22.40 |
| | 0.58 | 0.76 | 0.80 | 0.70 | 0.80 | 0.84 | 0.84 | 0.80 | 0.94 | 0.94 | 0.92 | 0.90 | 0.82 |
| | 15.93 | 19.53 | 23.23 | 19.88 | 18.38 | 22.83 | 21.07 | 22.88 | 25.28 | 26.39 | 26.68 | 26.68 | 22.40 |
| ViP-NeRF (Somraj & Soundararajan, 2023) | 0.34 | 0.18 | 0.26 | 0.32 | 0.32 | 0.28 | 0.22 | 0.22 | 0.09 | 0.11 | 0.12 | 0.12 | 0.22 |
| | 0.33 | 0.58 | 0.58 | 0.53 | 0.47 | 0.55 | 0.50 | 0.43 | 0.66 | 0.65 | 0.77 | 0.60 | 0.55 |
| | 12.97 | 16.58 | 18.63 | 16.12 | 14.82 | 16.25 | 14.14 | 18.04 | 17.67 | 14.75 | 20.85 | 18.65 | 16.62 |
| SimpleNeRF (Somraj et al., 2023) | 0.22 | 0.32 | 0.24 | 0.24 | 0.28 | 0.27 | 0.23 | 0.15 | 0.31 | 0.36 | 0.17 | 0.25 | 0.25 |
| | 0.74 | 0.68 | 0.74 | 0.75 | 0.75 | 0.77 | 0.79 | 0.90 | 0.77 | 0.67 | 0.84 | 0.81 | 0.77 |
| | 12.90 | 11.29 | 14.17 | 13.42 | 11.44 | 12.23 | 15.31 | 20.41 | 13.97 | 10.93 | 17.41 | 14.66 | 14.01 |
| VGOS (Sun et al., 2023) | 0.28 | 0.38 | 0.29 | 0.26 | 0.28 | 0.27 | 0.38 | 0.16 | 0.51 | 0.47 | 0.29 | 0.15 | 0.31 |
| | 0.69 | 0.65 | 0.71 | 0.76 | 0.74 | 0.76 | 0.62 | 0.90 | 0.58 | 0.58 | 0.75 | 0.87 | 0.72 |
| | 9.84 | 8.34 | 10.50 | 11.91 | 8.51 | 9.14 | 7.27 | 18.86 | 5.38 | 5.80 | 11.81 | 16.74 | 10.34 |
| SparseNeRF (Wang et al., 2023) | 0.23 | 0.12 | 0.15 | 0.37 | 0.14 | 0.14 | 0.12 | 0.14 | 0.04 | 0.04 | 0.11 | 0.08 | 0.14 |
| | 0.63 | 0.81 | 0.79 | 0.59 | 0.84 | 0.84 | 0.84 | 0.84 | 0.96 | 0.95 | 0.90 | 0.92 | 0.83 |
| | 17.14 | 21.11 | 24.88 | 12.36 | 22.25 | 23.05 | 20.85 | 19.75 | 27.52 | 28.98 | 23.74 | 28.00 | 22.47 |
| ZeroRF (Shi et al., 2024) | 0.45 | 0.36 | 0.41 | 0.45 | 0.29 | 0.30 | 0.33 | 0.27 | 0.19 | 0.19 | 0.24 | 0.30 | 0.31 |
| | 0.33 | 0.55 | 0.47 | 0.41 | 0.65 | 0.68 | 0.57 | 0.68 | 0.84 | 0.83 | 0.74 | 0.63 | 0.61 |
| | 11.55 | 12.43 | 11.81 | 12.84 | 15.66 | 16.01 | 12.77 | 16.50 | 17.81 | 15.34 | 16.64 | 14.25 | 14.47 |
| FrugalNeRF (Ours) | 0.19 | 0.14 | 0.18 | 0.22 | 0.21 | 0.13 | 0.13 | 0.12 | 0.06 | 0.05 | 0.10 | 0.11 | 0.14 |
| | 0.69 | 0.76 | 0.77 | 0.69 | 0.79 | 0.84 | 0.82 | 0.89 | 0.94 | 0.94 | 0.89 | 0.90 | 0.83 |
| | 17.38 | 19.06 | 22.38 | 18.96 | 17.77 | 24.01 | 20.35 | 26.11 | 24.57 | 25.85 | 25.43 | 27.28 | 22.43 |
| FrugalNeRF w/ mono. depth (Ours) | 0.19 | 0.13 | 0.17 | 0.21 | 0.20 | 0.13 | 0.13 | 0.12 | 0.06 | 0.05 | 0.08 | 0.10 | 0.13 |
| | 0.68 | 0.78 | 0.78 | 0.73 | 0.79 | 0.84 | 0.82 | 0.88 | 0.95 | 0.93 | 0.91 | 0.91 | 0.83 |
| | 17.14 | 19.89 | 23.17 | 20.33 | 17.18 | 23.71 | 20.59 | 26.60 | 25.52 | 25.04 | 27.84 | 27.10 | 22.84 |

Table 11: **Quantitative results on the DTU (Jensen et al., 2014) dataset with four input views. The three rows show LPIPS, SSIM and PSNR scores, respectively.**

| Method \ Scene | Scan21 | Scan31 | Scan34 | Scan38 | Scan40 | Scan41 | Scan45 | Scan55 | Scan63 | Scan82 | Scan103 | Scan114 | Average |
|---|---|---|---|---|---|---|---|---|---|---|---|---|---|
| FreeNeRF (Yang et al., 2023) | 0.18 | 0.14 | 0.13 | 0.24 | 0.14 | 0.12 | 0.09 | 0.06 | 0.04 | 0.03 | 0.08 | 0.07 | 0.11 |
| | 0.72 | 0.81 | 0.83 | 0.72 | 0.85 | 0.86 | 0.86 | 0.92 | 0.96 | 0.96 | 0.93 | 0.93 | 0.86 |
| | 18.72 | 21.29 | 25.97 | 19.43 | 22.88 | 25.59 | 22.39 | 28.63 | 27.35 | 31.51 | 27.30 | 28.65 | 24.98 |
| ViP-NeRF (Somraj & Soundararajan, 2023) | 0.33 | 0.19 | 0.21 | 0.31 | 0.35 | 0.24 | 0.23 | 0.24 | 0.08 | 0.08 | 0.10 | 0.12 | 0.21 |
| | 0.39 | 0.61 | 0.59 | 0.59 | 0.45 | 0.61 | 0.52 | 0.38 | 0.67 | 0.67 | 0.76 | 0.64 | 0.57 |
| | 14.24 | 17.22 | 19.44 | 18.19 | 15.76 | 18.84 | 15.57 | 16.62 | 17.19 | 16.45 | 22.67 | 19.50 | 17.64 |
| SimpleNeRF (Somraj et al., 2023) | 0.27 | 0.28 | 0.23 | 0.25 | 0.32 | 0.27 | 0.25 | 0.21 | 0.27 | 0.27 | 0.18 | 0.29 | 0.26 |
| | 0.71 | 0.73 | 0.78 | 0.75 | 0.72 | 0.76 | 0.78 | 0.88 | 0.82 | 0.80 | 0.84 | 0.81 | 0.78 |
| | 11.81 | 12.95 | 14.72 | 12.71 | 10.42 | 11.67 | 14.12 | 18.84 | 14.05 | 14.43 | 16.87 | 14.23 | 13.90 |
| VGOS (Sun et al., 2023) | 0.27 | 0.35 | 0.31 | 0.28 | 0.27 | 0.27 | 0.37 | 0.16 | 0.43 | 0.42 | 0.28 | 0.18 | 0.30 |
| | 0.73 | 0.69 | 0.71 | 0.74 | 0.76 | 0.78 | 0.64 | 0.90 | 0.66 | 0.66 | 0.75 | 0.85 | 0.74 |
| | 11.09 | 9.53 | 10.57 | 11.15 | 9.12 | 10.00 | 8.10 | 19.53 | 6.55 | 7.14 | 12.69 | 15.65 | 10.93 |
| SparseNeRF (Wang et al., 2023) | 0.16 | 0.14 | 0.15 | 0.21 | 0.21 | 0.14 | 0.10 | 0.09 | 0.04 | 0.05 | 0.09 | 0.06 | 0.12 |
| | 0.72 | 0.80 | 0.85 | 0.74 | 0.80 | 0.86 | 0.86 | 0.88 | 0.95 | 0.95 | 0.93 | 0.93 | 0.86 |
| | 18.60 | 20.99 | 25.87 | 20.92 | 19.45 | 24.81 | 22.15 | 26.37 | 26.20 | 26.72 | 28.10 | 28.19 | 24.03 |
| ZeroRF (Shi et al., 2024) | 0.43 | 0.32 | 0.28 | 0.44 | 0.28 | 0.25 | 0.20 | 0.29 | 0.17 | 0.14 | 0.26 | 0.32 | 0.28 |
| | 0.36 | 0.62 | 0.66 | 0.47 | 0.68 | 0.73 | 0.73 | 0.67 | 0.87 | 0.87 | 0.72 | 0.62 | 0.67 |
| | 11.75 | 13.48 | 16.47 | 13.53 | 16.87 | 17.26 | 16.48 | 15.92 | 19.33 | 19.12 | 15.18 | 13.36 | 15.73 |
| FrugalNeRF (Ours) | 0.17 | 0.12 | 0.16 | 0.17 | 0.19 | 0.12 | 0.12 | 0.12 | 0.05 | 0.04 | 0.07 | 0.10 | 0.12 |
| | 0.73 | 0.81 | 0.81 | 0.79 | 0.81 | 0.85 | 0.85 | 0.89 | 0.95 | 0.95 | 0.93 | 0.92 | 0.86 |
| | 19.21 | 21.84 | 24.99 | 23.08 | 19.47 | 25.64 | 21.59 | 27.31 | 26.27 | 27.26 | 29.27 | 28.21 | 24.51 |
| FrugalNeRF w/ mono. depth (Ours) | 0.17 | 0.12 | 0.15 | 0.17 | 0.19 | 0.12 | 0.11 | 0.12 | 0.05 | 0.03 | 0.07 | 0.09 | 0.12 |
| | 0.73 | 0.81 | 0.82 | 0.80 | 0.82 | 0.86 | 0.86 | 0.90 | 0.96 | 0.95 | 0.93 | 0.92 | 0.86 |
| | 19.07 | 21.65 | 25.82 | 23.13 | 18.96 | 25.55 | 22.21 | 28.02 | 26.87 | 28.28 | 29.27 | 28.92 | 24.81 |

Table 12: **Quantitative results on the RealEstate-10K Zhou et al. (2018) dataset.** For SimpleNeRF Somraj et al. (2023) and ViP-NeRF Somraj & Soundararajan (2023), we calculate metrics using testing data provided in their respective clouds. As for other models, we rely on the scores provided in the SimpleNeRF paper.

| Method | Venue | Learned priors | 2-view PSNR ↑ | 2-view SSIM ↑ | 2-view LPIPS ↓ | 3-view PSNR ↑ | 3-view SSIM ↑ | 3-view LPIPS ↓ | 4-view PSNR ↑ | 4-view SSIM ↑ | 4-view LPIPS ↓ | Training time ↓ |
|---|---|---|---|---|---|---|---|---|---|---|---|---|
| RegNeRF Niemeyer et al. (2022) | CVPR 2022 | normalizing flow | 16.87 | 0.59 | 0.45 | 17.73 | 0.61 | 0.44 | 18.25 | 0.62 | 0.44 | 2.35 hrs |
| DS-NeRF Deng et al. (2022) | CVPR 2022 | - | 25.44 | 0.79 | 0.32 | 25.94 | 0.79 | 0.32 | 26.28 | 0.79 | 0.33 | 3.5 hrs |
| DDP-NeRF Roessle et al. (2022) | CVPR 2022 | depth completion | 26.15 | 0.85 | 0.15 | 25.92 | 0.85 | 0.16 | 26.48 | 0.86 | 0.16 | 3.5 hrs |
| FreeNeRF Yang et al. (2023) | CVPR 2023 | - | 14.50 | 0.54 | 0.55 | 15.12 | 0.57 | 0.54 | 16.25 | 0.60 | 0.54 | 1.5 hrs |
| ViP-NeRF Somraj & Soundararajan (2023) | SIGGRAPH 2023 | - | 29.55 | 0.87 | 0.09 | 29.75 | 0.88 | 0.11 | 30.47 | 0.88 | 0.11 | 13.5 hrs |
| SimpleNeRF Somraj et al. (2023) | SIGGRAPH Asia 2023 | - | 30.30 | 0.88 | 0.07 | 31.40 | 0.89 | 0.08 | 31.73 | 0.89 | 0.09 | 9.5 hrs |
| FrugalNeRF (Ours) | - | - | 30.12 | 0.87 | 0.07 | 31.04 | 0.89 | 0.06 | 31.78 | 0.90 | 0.06 | 20 mins |

Table 13: **Quantitative results on the RealEstate-10K Zhou et al. (2018) dataset with two input views. The three rows show LPIPS, SSIM, and PSNR scores, respectively.**

| Scene / Method | 0 | 1 | 3 | 4 | 6 | Average |
|---|---|---|---|---|---|---|
| RegNeRF Niemeyer et al. (2022) | 0.35 | 0.32 | 0.49 | 0.54 | 0.54 | 0.45 |
| | 0.60 | 0.83 | 0.30 | 0.61 | 0.59 | 0.59 |
| | 16.51 | 21.04 | 13.88 | 17.13 | 15.79 | 16.87 |
| DS-NeRF Deng et al. (2022) | 0.26 | 0.27 | 0.51 | 0.24 | 0.31 | 0.32 |
| | 0.81 | 0.91 | 0.50 | 0.88 | 0.83 | 0.79 |
| | 24.68 | 27.93 | 19.24 | 29.18 | 26.18 | 25.44 |
| DDP-NeRF Roessle et al. (2022) | 0.11 | 0.12 | 0.34 | 0.06 | 0.11 | 0.15 |
| | 0.89 | 0.95 | 0.56 | 0.94 | 0.92 | 0.85 |
| | 25.90 | 25.87 | 18.97 | 32.01 | 28.00 | 26.15 |
| FreeNeRF Yang et al. (2023) | 0.45 | 0.50 | 0.64 | 0.67 | 0.48 | 0.55 |
| | 0.54 | 0.77 | 0.28 | 0.49 | 0.58 | 0.53 |
| | 15.00 | 17.00 | 12.15 | 12.84 | 15.50 | 14.50 |
| ViP-NeRF Somraj & Soundararajan (2023) | 0.05 | 0.05 | 0.22 | 0.04 | 0.08 | 0.09 |
| | 0.94 | 0.97 | 0.56 | 0.95 | 0.93 | 0.87 |
| | 30.41 | 32.03 | 18.96 | 34.74 | 31.61 | 29.55 |
| SimpleNeRF Somraj et al. (2023) | 0.04 | 0.04 | 0.21 | 0.03 | 0.05 | 0.07 |
| | 0.95 | 0.97 | 0.56 | 0.95 | 0.96 | 0.88 |
| | 31.89 | 33.8 | 18.65 | 34.93 | 32.24 | 30.30 |
| FrugalNeRF (Ours) | 0.04 | 0.04 | 0.20 | 0.04 | 0.05 | 0.07 |
| | 0.94 | 0.97 | 0.56 | 0.95 | 0.95 | 0.87 |
| | 30.13 | 34.69 | 18.35 | 35.00 | 32.45 | 30.12 |

Table 14: **Quantitative results on the RealEstate-10K Zhou et al. (2018) dataset with three input views. The three rows show LPIPS, SSIM, and PSNR scores, respectively.**

| Scene / Method | 0 | 1 | 3 | 4 | 6 | Average |
|---|---|---|---|---|---|---|
| RegNeRF Niemeyer et al. (2022) | 0.40 | 0.32 | 0.53 | 0.56 | 0.37 | 0.44 |
| | 0.60 | 0.82 | 0.29 | 0.62 | 0.71 | 0.61 |
| | 15.99 | 20.89 | 13.87 | 17.60 | 20.28 | 17.73 |
| DS-NeRF Deng et al. (2022) | 0.24 | 0.26 | 0.53 | 0.26 | 0.31 | 0.32 |
| | 0.83 | 0.91 | 0.49 | 0.87 | 0.85 | 0.79 |
| | 25.24 | 28.68 | 19.14 | 29.08 | 27.58 | 25.94 |
| DDP-NeRF Roessle et al. (2022) | 0.11 | 0.11 | 0.38 | 0.06 | 0.13 | 0.16 |
| | 0.89 | 0.96 | 0.55 | 0.94 | 0.92 | 0.85 |
| | 25.27 | 26.67 | 18.81 | 31.84 | 26.99 | 25.92 |
| FreeNeRF Yang et al. (2023) | 0.54 | 0.51 | 0.64 | 0.59 | 0.42 | 0.54 |
| | 0.53 | 0.75 | 0.29 | 0.61 | 0.66 | 0.57 |
| | 13.79 | 15.59 | 12.45 | 15.72 | 18.05 | 15.12 |
| ViP-NeRF Somraj & Soundararajan (2023) | 0.06 | 0.10 | 0.26 | 0.04 | 0.08 | 0.11 |
| | 0.94 | 0.95 | 0.60 | 0.95 | 0.95 | 0.88 |
| | 30.66 | 29.89 | 19.59 | 35.17 | 33.43 | 29.75 |
| SimpleNeRF Somraj et al. (2023) | 0.04 | 0.04 | 0.23 | 0.03 | 0.08 | 0.08 |
| | 0.95 | 0.98 | 0.61 | 0.95 | 0.95 | 0.89 |
| | 32.23 | 36.44 | 19.65 | 35.85 | 32.81 | 31.40 |
| FrugalNeRF (Ours) | 0.04 | 0.03 | 0.18 | 0.03 | 0.04 | 0.06 |
| | 0.95 | 0.98 | 0.61 | 0.95 | 0.96 | 0.89 |
| | 31.11 | 35.39 | 18.85 | 35.78 | 34.07 | 31.04 |

Table 15: **Quantitative results on the RealEstate-10K Zhou et al. (2018) dataset with four input views. The three rows show LPIPS, SSIM, and PSNR scores, respectively.**

| Scene / Method | 0 | 1 | 3 | 4 | 6 | Average |
|---|---|---|---|---|---|---|
| RegNeRF Niemeyer et al. (2022) | 0.43 | 0.35 | 0.59 | 0.56 | 0.27 | 0.44 |
| | 0.59 | 0.83 | 0.29 | 0.65 | 0.75 | 0.62 |
| | 16.09 | 20.98 | 13.91 | 18.48 | 21.78 | 18.25 |
| DS-NeRF Deng et al. (2022) | 0.27 | 0.26 | 0.56 | 0.25 | 0.31 | 0.33 |
| | 0.82 | 0.92 | 0.50 | 0.87 | 0.85 | 0.79 |
| | 25.40 | 29.40 | 19.64 | 29.26 | 27.69 | 26.28 |
| DDP-NeRF Roessle et al. (2022) | 0.12 | 0.08 | 0.39 | 0.06 | 0.13 | 0.16 |
| | 0.89 | 0.96 | 0.58 | 0.93 | 0.91 | 0.86 |
| | 25.14 | 28.57 | 19.57 | 31.73 | 27.36 | 26.48 |
| FreeNeRF Yang et al. (2023) | 0.56 | 0.48 | 0.65 | 0.58 | 0.39 | 0.53 |
| | 0.53 | 0.80 | 0.31 | 0.66 | 0.69 | 0.60 |
| | 13.84 | 17.93 | 12.69 | 17.29 | 19.48 | 16.25 |
| ViP-NeRF Somraj & Soundararajan (2023) | 0.06 | 0.08 | 0.27 | 0.05 | 0.09 | 0.11 |
| | 0.94 | 0.96 | 0.62 | 0.94 | 0.95 | 0.88 |
| | 31.64 | 32.24 | 20.35 | 34.84 | 33.28 | 30.47 |
| SimpleNeRF Somraj et al. (2023) | 0.04 | 0.05 | 0.24 | 0.03 | 0.09 | 0.09 |
| | 0.96 | 0.97 | 0.64 | 0.95 | 0.94 | 0.89 |
| | 32.95 | 36.44 | 20.52 | 35.97 | 32.77 | 31.73 |
| FrugalNeRF (Ours) | 0.04 | 0.03 | 0.17 | 0.03 | 0.05 | 0.06 |
| | 0.96 | 0.98 | 0.64 | 0.95 | 0.96 | 0.90 |
| | 32.29 | 36.06 | 19.81 | 36.54 | 34.22 | 31.78 |

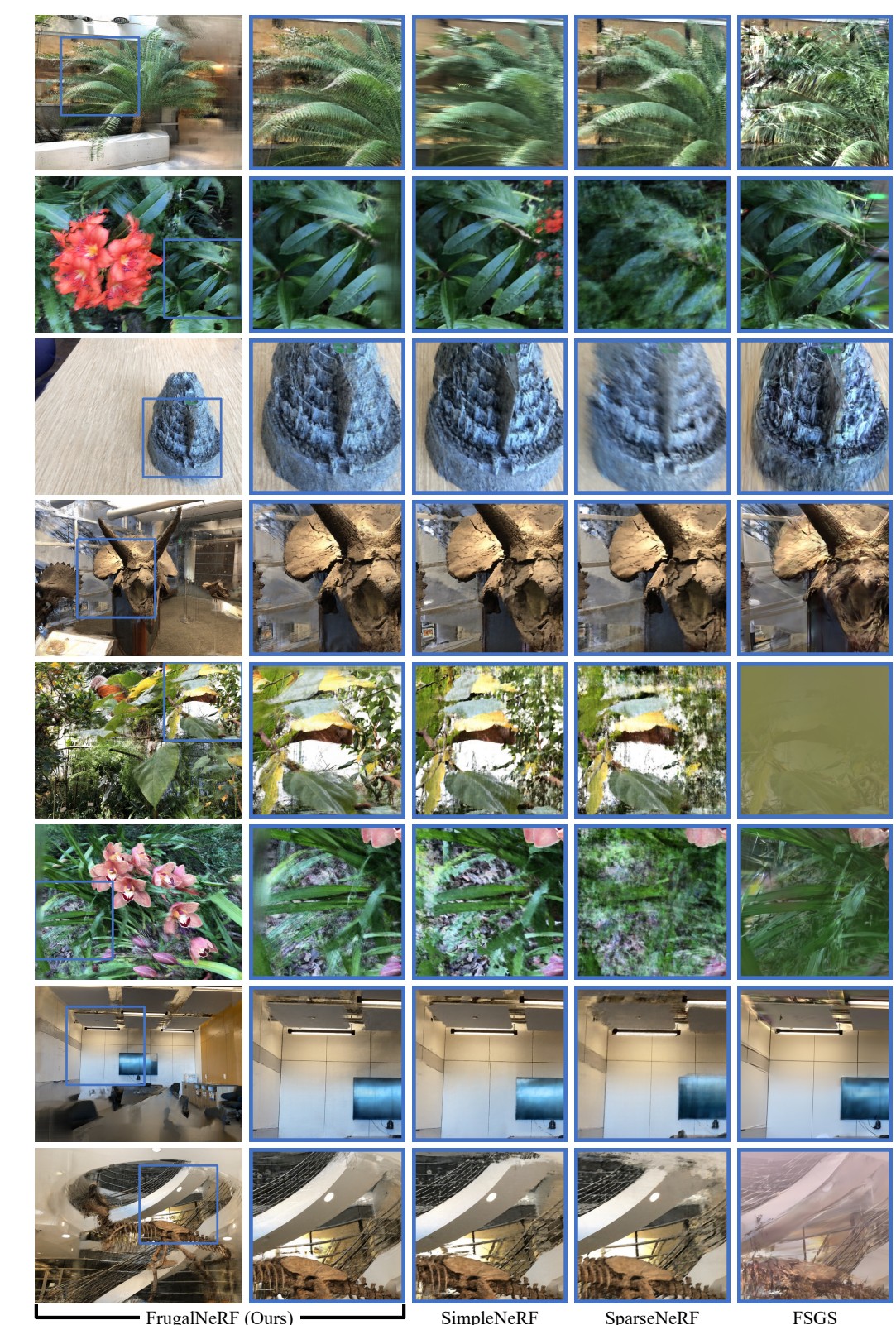

Figure 10: **More qualitative comparisons on the LLFF (Mildenhall et al., 2019a) dataset with two input views.** FrugalNeRF achieves better synthesis quality in different scenes.

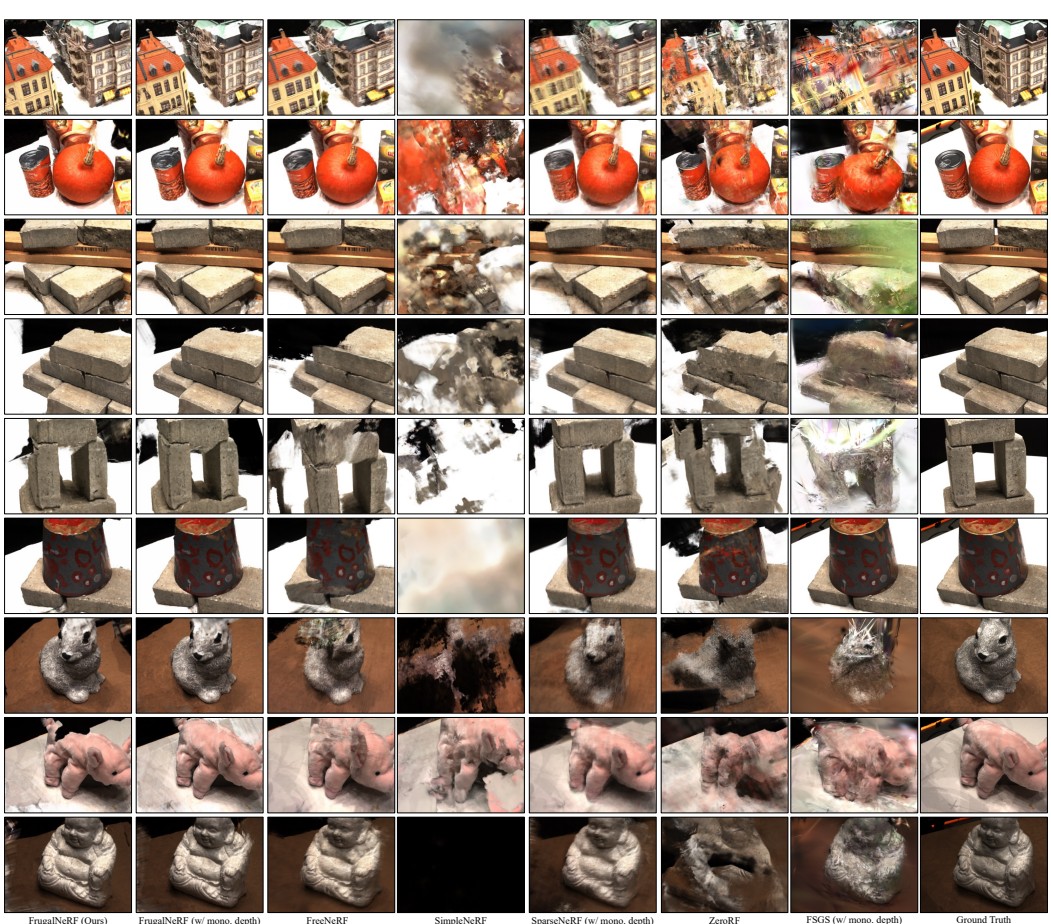

Figure 11: **More qualitative comparisons on the DTU (Jensen et al., 2014) dataset with two input views.** FrugalNeRF achieves better synthesis quality in different scenes.

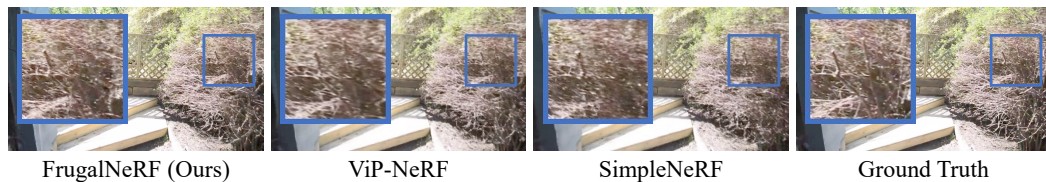

Figure 12: **Qualitative comparisons on the RealEstate-10K Zhou et al. (2018) dataset with two input views.** Compared to Vip-NeRF Somraj & Soundararajan (2023) and SimpleNeRF Somraj et al. (2023), our FrugalNeRF renders sharper details in the scene.

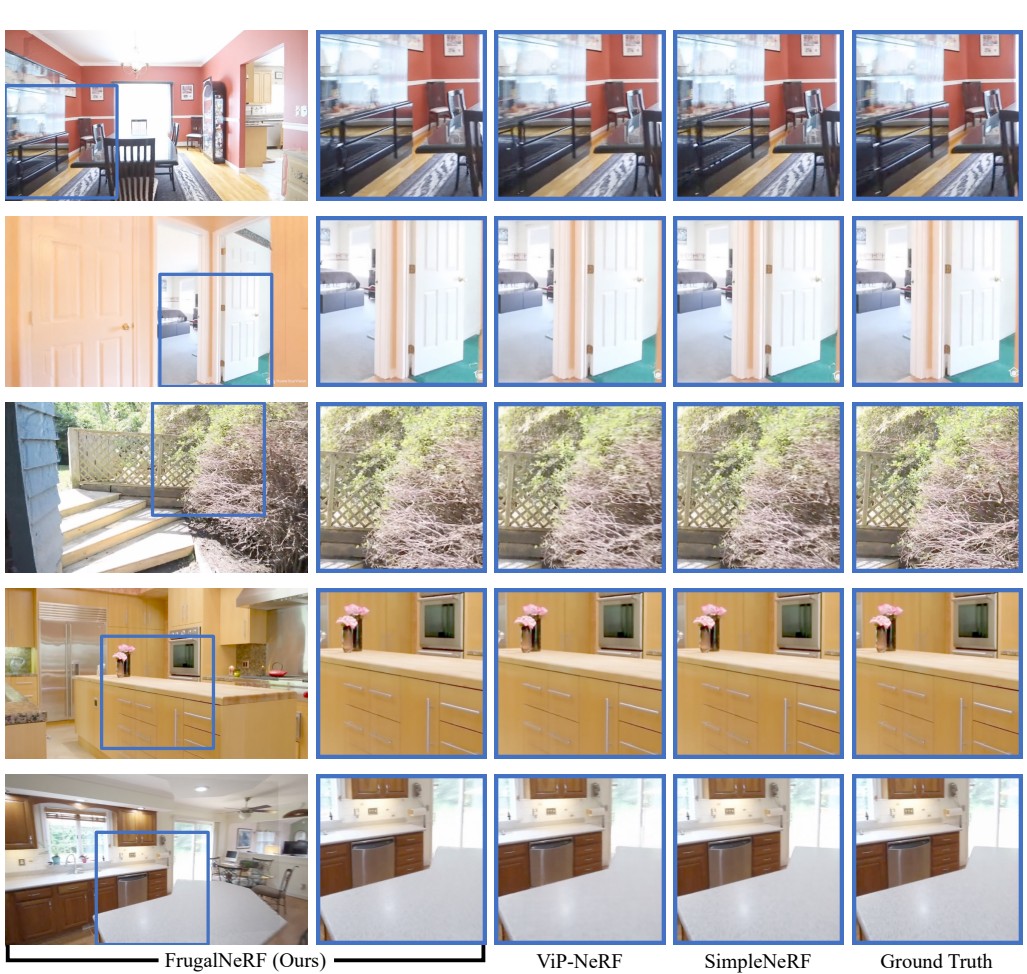

Figure 13: **More qualitative comparisons on the RealEstate-10K Zhou et al. (2018) dataset with two input views.** FrugalNeRF achieves synthesis quality comparable to the state-of-the-art methods.

