# OpenReview forum: "FrugalNeRF: Fast Convergence for Few-shot Novel View Synthesis without Learned Priors"
_ICLR.cc/2025/Conference — ICLR 2025 Conference Withdrawn Submission_

### Official Review · Reviewer_ynSY · 2024-10-29

**Soundness:** 2
**Presentation:** 2
**Contribution:** 2
**Rating:** 5
**Confidence:** 4

**Summary:**

This paper introduces a few-shot NeRF framework that leverages weight-sharing voxels across multiple scales to represent scene details efficiently. Experiments demonstrate incremental results across various datasets.

**Strengths:**

Advantages:
Novelty: The primary contributions of this work lie in two areas:
A weight-sharing voxel representation that encodes multiple frequency components within the scene, enhancing efficiency in scene representation.
A geometric adaptation technique that selects accurate rendered depth across scales via reprojection errors, creating pseudo-ground-truth depth to guide the training process.

**Weaknesses:**

Drawbacks:
Limited Novelty: The proposed contributions are not entirely novel.
For the multi-frequency component, prior work, such as FreeNeRF, has addressed similar challenges by gradually adding frequency.
For geometric adaptation, many existing few-shot novel view synthesis methods, such as SparseNeRF, already utilize projections to refine geometry.
As a result, it is challenging to identify a clear, distinct novelty in this paper.

**Questions:**

Comparison with FreeNeRF: How does FrugalNeRF differ from FreeNeRF’s frequency-based approach?

---

### Official Review · Reviewer_Cgr3 · 2024-11-03

**Soundness:** 3
**Presentation:** 3
**Contribution:** 2
**Rating:** 5
**Confidence:** 3

**Summary:**

This paper introduces a fast few-shot Nerf for reconstructing scenes without extra priors. The core idea is to leverage the reprojection errors between different scales of voxel to select pseudo depth for reliable supervision. The quantitative evaluation surpasses existing few-shot Nerf while reducing the training times.

**Strengths:**

1. The overall framework is sound. The proposed weight-sharing voxel representation indeed encapsulates scene components from different frequencies, which reasonably motivates the following cross-scale geometric adaptation.
2. Good presentation. The illustration is clear and easy to follow.
3. The authors provide sufficient extra information in the supplementary, which helps more comprehensively understand this article.

**Weaknesses:**

1. One of the keywords in this paper is "fast", but as far as I know, there are several approaches in few-shot Nerf that focus on faster training, e.g. VGOS[1] and DNGaussian[2]. It seems that the authors omit the comparisons in the main tables/figures with these methods. I think these comparisons are important. For VGOS, it is a voxel-based method without using explicit priors from other models, which is most similar to the setting claim by this paper. For DNGaussian, although it uses explicit outside depth priors, its training speed is still super-fast as it only costs 3.5 minutes to train the 3-view reconstruction in LLFF, where the proposed method needs 6 minutes even without the multi-scale voxel representations (Table 3), let alone the situation when L>0.
2. The proposed method of this paper tends to produce much more high-frequency artifacts, most scenes exhibit a distinct hierarchy of objects when the camera starts spinning around, making the visualization look worse. Does it suggest that the proposed cross-scale geometric adaption is more likely to produce inconsistent geometric? The original claim is that selecting pseudo depth according to reprojection errors helps improve geometric consistency, a few explanations would be better.
4. Moreover, I didn't see any correction strategy once the pseudo depths were inaccurate. Since the supervisions of pseudo depth are imbalanced across different scales, it is confusing how to reorganize different frequency components coherently into the weight-sharing voxel without introducing high-frequency artifacts.
5. I wonder if the authors could evaluate their method on the Realistic Synthetic 360. Since it is a 360 surrounding dataset, which is more suitable for evaluating the robustness of the proposed method against occlusions.


[1] Sun J, Zhang Z, Chen J, et al. Vgos: Voxel grid optimization for view synthesis from sparse inputs[J]. arXiv preprint arXiv:2304.13386, 2023.
[2] Li J, Zhang J, Bai X, et al. Dngaussian: Optimizing sparse-view 3d gaussian radiance fields with global-local depth normalization[C]//Proceedings of the IEEE/CVF Conference on Computer Vision and Pattern Recognition. 2024: 20775-20785.

**Questions:**

Please refer to the weaknesses.

---

### Official Review · Reviewer_A5QU · 2024-11-04

**Soundness:** 2
**Presentation:** 2
**Contribution:** 2
**Rating:** 5
**Confidence:** 4

**Summary:**

This paper introduces a NeRF-based few-shot novel view synthesis without relying on externally learned priors. The novelty of this work mainly lies in: 1) the multi-scales weight-sharing voxels for scene representation; 2) the depth supervision in training by selecting accurate rendered depth across different scales based on the reprojection error from both training and novel views. The approach demonstrates performance improvements on several standard benchmarks including LLFF and DTU.

**Strengths:**

This paper introduces a NeRF-based few-shot novel view synthesis with cross-scale geometric adaptation training scheme, which works well in few-shot scenarios and does not rely on externally learned priors.  The strengths of the work mainly lies in: 1) the multi-scales weight-sharing voxels for scene representation; 2) the depth supervision in training by selecting accurate rendered depth across different scales based on the reprojection error from both training and novel views. 3) No pre-trained models involved and remarkable training time reduction.

**Weaknesses:**

1. The main concern is the novelty of this work is limited. Concretely, utilizing pseudo-depth as training supervision in NeRF is not new. Also, utilizing pseudo-depth in NeRF without externally learned depth priors is also not new [1]. Moreover, the novelty of multi-scales voxels [1,2] and cross-view warping [3,4] is limited. It is suggested to compare with the related approaches and illustrate the novelty of the proposed approach.
2. The experimental results in Table 4 show that the performance improvement from the novel views is quite limited (17.84 vs 18.07) compared with other components. Please explain it more.

[1] Li, J., Zhou, Q., Yu, C., Lu, Z., Xiao, J., Wang, Z., & Wang, F. (2023). Improved Neural Radiance Fields Using Pseudo-depth and Fusion. ACM Symposium on Neural Gaze Detection.2023
[2] Thomas M¨uller, Alex Evans, Christoph Schied, and Alexander Keller. Instant neural graphics primitives with a multiresolution hash encoding. ACM Transactions on Graphics (ToG), 2022.
[3] Ahn, Young Chun et al. PANeRF: Pseudo-view Augmentation for Improved Neural Radiance Fields Based on Few-shot Inputs. ArXiv abs/2211.12758 (2022): n. pag.
[4] Yan, D., Huang, G., Quan, F., & Chen, H. (2024). MSI-NeRF: Linking Omni-Depth with View Synthesis through Multi-Sphere Image aided Generalizable Neural Radiance Field. ArXiv, abs/2403.10840.

**Questions:**

see weaknesses

---

### Official Review · Reviewer_8hgo · 2024-11-07

**Soundness:** 2
**Presentation:** 3
**Contribution:** 2
**Rating:** 5
**Confidence:** 4

**Summary:**

The paper introduces a novel approach to accelerate NeRF training in few-shot scenarios without relying on external pretrained priors. The method employs a voxel-based representation for both density and appearance. The main contribution lies in the introduction of a cross-scale geometric loss. Specifically, voxel grids are downsampled at multiple scales, and for each pixel, the rendered depth at each scale is supervised using the depth value from the scale with the lowest reprojection error at that pixel. The approach is evaluated on the LLFF, DTU, and RealEstate-10K benchmarks, demonstrating performance on par with state-of-the-art methods while achieving faster training times.

**Strengths:**

- The paper is clearly written and easy to follow.
- The evaluations are comprehensive, demonstrating competitive performance relative to state-of-the-art methods with a reduced training duration.

**Weaknesses:**

- **Motivation and Theoretical Justification**: The paper lacks a strong motivation and theoretical basis for the proposed regularization. Specifically, why are the colors at all scales supervised using high-frequency ground-truth color values? Regarding depth supervision, if, early in training, the coarse levels yield the lowest reprojection errors, the pseudo ground-truth depth is derived from these coarse levels. In such cases, how can the depth predictions of finer levels acquire more detailed features if they are constrained by the coarser levels' current outputs?

- **Comparisons with Recent Methods**: The paper does not include comparisons with recent state-of-the-art methods such as SparseCraft[1], MixNeRF[2], and FlipNeRF[3]. Furthermore, it would be beneficial to present results using the (more) standard setting" with 3, 6, and 9 views, as adopted by prior works (e.g., RegNeRF , FreeNeRF, SparseCraft [1], MixNeRF [2], FlipNeRF [3]), unless a compelling justification is provided for choosing a different evaluation setting.

[1] SparseCraft: Few-Shot Neural Reconstruction through Stereopsis Guided Geometric Linearization.  ECCV24.

[2] MixNeRF: Modeling a Ray with Mixture Density for Novel View Synthesis from Sparse Inputs. CVPR23.

[3]FlipNeRF: Flipped Reflection Rays for Few-shot Novel View Synthesis. ICCV23.

**Questions:**

-  What's the effect of the cross-scale depth supervision loss when a monocular depth prior is used ?

---

### Official Review · Reviewer_7fEp · 2024-11-08

**Soundness:** 3
**Presentation:** 3
**Contribution:** 2
**Rating:** 5
**Confidence:** 4

**Summary:**

This paper presents a few-shot NeRF framework FrugalNeRF, which leverages weight-sharing voxels across multiple scales to represent various scene details efficiently. The main contribution is a across-scale geometric adaptation training scheme, which selects pseudo  ground-truth depth based on reprojection error from both training and novel views across scales. Extensive experiments on various datasets show the effectiveness of the proposed method.

**Strengths:**

1.	This paper is easy to follow and well-written.
2.	This paper introduces a weight-sharing voxel representation that encodes multiple frequency components of the scene, which enhances the efficiency and quality of few-shot novel view synthesis.
3.	This paper proposes a cross-scale geometric adaptation strategy which enables a robust learning mechanism that is less reliant on complex scheduling and more adaptable to various scenes.

**Weaknesses:**

1.	The contributions of this paper are incremental, introducing the voxel/TensoRF for fast training and convergence is a common strategy in NeRF, self-supervised consistency and frequency regularization also are widely used in few-shot NeRF. For example, ReVoRF[1]  explores pseudo-views unreliability within few-shot radiance fields to enhanced multi-view consistency learning with a bilateral
 geometric consistency loss and introduces the voxel-based representation to achieve fast training.
[1] Yingjie Xu, Bangzhen Liu, Hao Tang, Bailin Deng, Shengfeng He. Learning with Unreliability: Fast Few-shot Voxel Radiance Fields
 with Relative Geometric Consistency. CVPR 2024.
2.	The proposed method needs sparse prior obtained from COLMAP, however, COLMAP can not always run successfully in the setting of few-shot NeRF, such as the case in some scenes of DTU.  How the proposed method handles cases where COLMAP fails?
3.	As the number of views increases, the performance gap between FrugalNeRF and other methods, such as FreeNeRF and SparseNeRF, narrows. Therefore, how does the proposed method perform with 6 and 9 views, which are also common settings in few-shot NeRF? More quantitative results for 6 and 9 views are needed.

**Questions:**

Please see the Weaknesses.

---

### Note · Authors · 2024-11-14

I have read and agree with the venue's withdrawal policy on behalf of myself and my co-authors.